# Predicting Popularity of Video Streaming Services with Representation Learning: A Survey and a Real-World Case Study

**DOI:** 10.3390/s21217328

**Published:** 2021-11-03

**Authors:** Sidney Loyola de Sá, Antonio A. de A. Rocha, Aline Paes

**Affiliations:** Instituto de Computação, Universidade Federal Fluminense, Niterói 24210-330, RJ, Brazil; sidney_loyola@id.uff.br (S.L.d.S.); alinepaes@ic.uff.br (A.P.)

**Keywords:** popularity prediction, video, machine learning, word embeddings

## Abstract

The Internet’s popularization has increased the amount of content produced and consumed on the web. To take advantage of this new market, major content producers such as Netflix and Amazon Prime have emerged, focusing on video streaming services. However, despite the large number and diversity of videos made available by these content providers, few of them attract the attention of most users. For example, in the data explored in this article, only 6% of the most popular videos account for 85% of total views. Finding out in advance which videos will be popular is not trivial, especially given many influencing variables. Nevertheless, a tool with this ability would be of great value to help dimension network infrastructure and properly recommend new content to users. In this way, this manuscript examines the machine learning-based approaches that have been proposed to solve the prediction of web content popularity. To this end, we first survey the literature and elaborate a taxonomy that classifies models according to predictive features and describes state-of-the-art features and techniques used to solve this task. While analyzing previous works, we saw an opportunity to use textual features for video prediction. Thus, additionally, we propose a case study that combines features acquired through attribute engineering and word embedding to predict the popularity of a video. The first approach is based on predictive attributes defined by resource engineering. The second takes advantage of word embeddings from video descriptions and titles. We experimented with the proposed techniques in a set of videos from GloboPlay, the largest provider of video streaming services in Latin America. A combination of engineering features and embeddings using the Random Forest algorithm achieved the best result, with an accuracy of 87%.

## 1. Introduction

The Internet has become one of the primary means of communication and information in the world. To give an idea, in 2012, 2 billion people had access to the Internet, representing 30% of the world population [1]. Almost ten years later, the number of Internet users has grown to 4.66 billion, representing 60% of the world population driven by the increase in the use of smartphones and other smart devices [2]. Recently, the challenges imposed by COVID-19 were responsible for almost 300 million people to access the Internet for the first time in the last year, according to the DataReportal [3] website.

With the popularization of the Internet, streaming video services, such as YouTube, Netflix, GloboPlay, and Amazon Prime, have also grown. In April 2021, Netflix had 208 million subscribers while Amazon Prime had 200 million subscribers [4] worldwide. It is estimated that, in Brazil, there are 19.88 million subscribers to Video Streaming services [5], 6.5 million of whom are GloboPlay subscribers [6]. The GloboPlay Streaming service was launched in 2015 and, nowadays, it is the largest one in Latin America. In an interview with GloboPlay’s Digital Director, Erick Bretas stated, without revealing numbers that they ended the year 2020 with an 89% increase in subscribers. In addition, he said, “We are broadcasting 100 million hours every month” [7]. These services have complex and robust structures to deliver thousands of videos to millions of users.

Despite the massive amount of material offered by content providers, few videos attract the most attention from users. Thus, the ability to predict Web content’s popularity finds several practical applications for content producers, marketing, and infrastructure providers. To mention a few, the advantages of correctly predicting which content will become popular include: increasing the return on marketing investment [8], proactively allocating network resources by adjusting them accurately to future demands [9], selecting the best content for the audience [10,11], directing investments to the content to be produced [8], and increasing the hit rate of cache relocation algorithms [12].

However, finding out which videos will be popular in advance is not trivial. Several variables can directly influence the popularity, including the topics covered in the material, the attachment of the content to what is going on in the world, the lexical content, the linguistic or visual style, the authors of the content, the target audience, the content’s authors, among others. Because of so many influencing factors, researchers developed several strategies to improve the prediction’s performance. Among them, Artificial Intelligence (AI) techniques that can find patterns relating the content and its variables to the popularity have obtained successful results lately. Mainly, methods of Machine Learning (ML), aided by Natural Language Processing (NLP) when one has textual content, are the subareas of AI mostly prominent to that task [10,13,14,15,16].

ML aims at creating models that learn to solve a task from experience [17]. The experience is usually represented by a dataset gathered from the task [18]. In popularity forecast, the task can be designed as a classification or as a regression task, according to the prediction’s final objective. In order to handle the textual content, ML methods require that they are transformed into a categorical or numerical representation. Concerning the popularity prediction task, NLP allows for designing linguistic-based features or discovering them directly from the content [10,16]. Recently, NLP techniques that transform written and spoken content into a vector representation embedded into a vector space have become the standard practice [19,20].

Predicting content popularity on the Web has been researched by several authors who have already examined different types of content such as tweets [13,14], images [21], and videos [9,22,23]. One of the steps that most influences the outcome of predictive models is to define the predictive attributes. Motivated by that, in this manuscript, we identify the main methods used, their respective features, and the context in which the researchers applied them, facilitating the attribute engineering stage to use the popularity prediction models. Combining several features can improve the performance of the models already proposed according to the applied context. There is still no clear standardization in the literature in this regard, as identified by Zhou et al. [24]. Thus, we intend to evolve this discussion on feature combination by presenting a case study that combines features acquired through attribute engineering and word embeddings, both obtained from the title and description of videos of a streaming service.

We propose two approaches aiming at predicting video popularity from a streaming service. Both focus on the textual content of the videos (title and description). The first approach focuses on feature engineering to select relevant predictive features that are yielded from NLP methods. The second approach leverages representation learning techniques to obtain latent features automatically through word embeddings. We extract the features to learn six ML models to classify which videos will become popular. The ML classifiers are evaluated with quantitative metrics, namely Precision, Recall, F1-Score, and Accuracy. We investigate the predictive power of each classifier when they are induced from engineered features, word embeddings, and when both types of those features are at their disposal on a set of 9989 videos from GloboPlay’s streaming service. From the results, we found out that the best model was the Random Forest when using the dataset of the titles’ word embeddings concatenated with the features obtained with NLP techniques, reaching an accuracy of 87%.

In 2014, Tatar et al. [8] presented a survey on the main popularity prediction research, specifying a taxonomy focusing on the objective and timing of prediction execution: classification or regression and before or after the publication of the content. Recently, Moniz and Torgo [25] prepared a review of predictive models proposing a classification focused on three elements: objective, selection of predictive attributes, and methods of data mining/machine learning. In 2021, Zhou et al. [24] presented a study on popularity prediction, focusing on information dissemination and including scientific articles as one of the types of content to be studied. This manuscript follows a different approach compared to the previous surveys about the popularity prediction theme: given the plethora of possible variables and the multitude of existing ML algorithms employable to the problem, here we take a representation-based approach focusing on the attributes and how they are used for each ML method. Another contribution over the previous works is the description of the use of Deep Learning techniques to extract attributes directly from the videos’ frames, further extending to selecting attributes. In summary, the contributions of this work are:
A review of state-of-the-art popularity prediction methods focused on extracting attributes directly from the content of news articles, images, and videos.A taxonomy that classifies the models through the use of predictive features.Inclusion of recent studies that obtain predictive attributes directly from video frames.Analysis of the best predictive models for each work, since previous works sometimes present more than one approach.Implementation of predictive models with Python code publicly available [26].Extraction of Features from Titles and Descriptions of Globoplay’s videos.Comparison of models predictive using NLP and word embeddings.Evaluate popularity prediction of Globoplay’s videos using ML algorithms.

The rest of the manuscript is divided as follows: Section 2 introduces basic concepts of Machine Learning and Natural Language Processing, Section 3 presents the concepts of Popularity Prediction, its operation, types of content and taxonomy, in Section 4, the main Classification methods found in the literature are presented, in Section 5, the Regression methods are presented, in Section 6, we present the case study, identifying the methodology used, in Section 7, the results of experiments are presented. Finally, in Section 9, the conclusions are presented.

## 2. Theoretical Foundation

This section presents the basic concepts related to NLP. We briefly describe some models of representation and techniques used to extract attributes from the content. This entire section reflects the point of view of the task explored in this manuscript, namely, predicting the popularity of web content.

### 2.1. Machine Learning

Machine Learning is a subfield of AI that aims at equipping machines with the ability to solve problems that require learning from experience. The main motivation is that not every problem can be modeled and solved using a deterministic algorithm, which follows a step-by-step fashion. For example, recognizing people from their face, despite being a simple task for humans, is not trivial for a machine. The many variables involved in the problem make it difficult to implement with a standard algorithm. In front of these situations, ML techniques build knowledge without being “programmed” to do so but instead by “learning” (improve performance at some task) through examples [17].

In an ML technique, the learning is, in most of the times, to search for a target function capable of solving the problem to be addressed. By using previous data related to the task (the experience), the algorithms induce functions capable of achieving a particular objective by themselves. The experience is commonly referred as the dataset and is composed of examples (an individual experience) and attributes (variables describing the experience).

Below, we present some common definitions in ML applied to the context of popularity prediction:

**Dataset**. When dealing with attribute-value scenarios, the dataset is a tabular representation of the attributes that represent the studied objects [17]. In our case, it means web content.

**Feature**. Characteristic of the content, obtained directly or derived (through some calculation or technique). Each attribute is associated with an object property (Web content) [17].

**Predictive attributes**. These are features used as inputs for ML models. Usually, the entry is represented by an attribute vector [17].

**Target attribute**. In addition, called output, it represents the phenomenon of interest of the prediction, in our case, the popularity measure.

Each ML approach may choose a number of different strategies to learn the target function. This includes the representation of the experience, including matrices of examples and attributes, pairs of input and output or only inputs, interaction with the environment; the representation of the learned function, for example, functions, rules, probability distributions; and the way the method traverses the search space to find an approximation of the target function [17].

Regarding the type of gathered experience, ML methods follow three standard paradigms: supervised learning, unsupervised learning, and reinforcement learning (Other types of supervision also exist, namely, semi-supervised learning, when only a subset of the examples have an output; and self-supervised leaning, when the label is extracted from the task itself without human supervision.) In this manuscript, we focus on supervised techniques defined as follows.

**Supervised Learning**. In this paradigm, the tasks are predictive, and the training dataset must have input and output attributes. The output attributes are also called target variable. The outputs are labeled simulating the activity of a supervisor, that is, someone who knows the “answer”. The supervised learning task can be described as [18]:

Given a training set of *n* input and output pairs of examples
$$\left(x_{1},y_{1}\right),\left(x_{2},y_{2}\right),\cdots ,\left(x_{n},y_{n}\right),$$
where each $x_{i}$ is a set of attributes valued according to the example *i* and each $y_{i}$ was generated by an unknown function y=f(x). Thus, the problem is to find a function *h* that approximates the true function *f*.

The hypothesis function *h* must be valid for other objects in the same domain that do not belong to the training set. This property is called generalization. The low capacity for generalization means that the data are over-adjusted to the training set (overfitting) or under-adjusted to the data (underfitting) [17].

To measure the generalization capabilities, it is a common practice to adopt three sets of data: training, validation, and testing. The training set is used to learn the hypothesis function *h* from the examples. The validation set is important to verify if the model is neither over-adjusted nor under-adjusted. Finally, with the test set, the performance of the model is assessed, verifying whether it solves the proposed problem or not.

Predictive tasks are divided into **classification** or **regression**. In the former, the output is a set of discrete values, for example, the health status of a patient (*healthy*, *sick*). In the latter, the output is a numerical value, e.g.,: *temperature*. Russell and Norvig [18] present the following definitions:
Classification: $y_{i}=f\left(x_{i}\right)\;\in \;\left\{c_{1},\cdots ,c_{m}\right\}$, that is, $f(x_{i})$ accepts values in a discrete and unordered set;Regression: $y_{i}f(x_{i})\;\in \;R$ that is, $f(x_{i})$ accepts values in an infinite and ordered set.

### 2.2. Natural Language Processing

Natural Language Processing is a subfield in the intersection of AI and Computational Linguistics that investigates methods and techniques through which computational agents can communicate with humans. Among the various existing communication formats, what interests us is writing because the Web, our study context, registers a large part of human knowledge through innumerable information pages. Computers use formal languages, such as Java or Python programming languages, with sentences precisely defined by a syntax, verifying whether a set of strings is valid or not in a given language. On the other hand, humans use ambiguous and confusing communication.

There are two commonly used strategies to extract features from texts to feed ML methods. One way is manually engineering features based on linguistic cues and experts’ experience and compute values to those features from the texts. The other way is representing the texts into a vector space relying on the distributional semantics [27]. In this case, two approaches are possible. The first one defines the features as the words in the vocabulary, and the values are measured based on the frequency of the words in the example. This is known as bag-of-words. The other strategy induces a language model from a large set of texts, relying on a probabilistic or a neural formulation [28,29].

Language models can be induced from characters, the basic unit, words, sentences, and documents. We will illustrate a language model from characters. The probability distribution over strings is commonly written as $P(c_{1:n})$. Using these probabilities, we can create models defined as a **Markov chain** of order n−1. In these chains, the probability of the character $c_{i}$ depends on the immediately preceding characters. Thus, given a sequence of characters, we can estimate what will be the next character. We call these stripe sets of probabilities of n-gram models. In Equation (Equation 1), we have a trigram model (3-gram) [28]. These models do not need to be restricted to sets of characters; they can be extended to word sets:
(1)$$P\left(c_{i}|c_{1:i-1}\right)=P\left(c_{i}|c_{i-2:i-1}\right)$$

The *bag*-*of*-*words* formulation does not take into account the order of the words. In addition, there is no capture of semantic values. All words have the same importance, differing from each other only by their frequency. This model can be extended to use the n-grams previously presented, counting the set of *n* words.

Tasks and methods are built upon the *bag*-*of*-*words* formulation. A popular task is sentiment analysis to classify the texts according to their polarity, negative, positive, or neutral. In this sense, the use of *bag*-*of*-*words* with the SVM classifier is one of the most efficient models to classify a text as positive or negative, as seen in Agarwal and Mittal [30]. A popular method is *Latent Dirichlet Alocation* (LDA) to find topics into texts. LDA is a probabilistic model representing the corpus at three levels: topics, documents, and words. The topics are separated according to their frequencies through the concept of *bag*-*of*-*words* [31].

Several NLP tasks can be addressed with language models. We can mention named entity recognition (NER), recognition of handwritten texts [32], language recognition, spelling correction, and gender classification [18]. The recognition of named entities uses several techniques. One of the simplest is to find sequences that allow the identification of people, places, or Organizations. For example, the strings “Mr”, “Mrs”, “Dr” make it possible to identify people; in addition, “street” and “Av”, make it possible to identify places. These n-gram models can locate more complex entities as demonstrated in Downey et al. [33]. Much of the work presented in this article uses the Stanford NER [34], a JAVA implementation of a NER recognizer. This software is already pre-trained to recognize people, organizations, and places in the English language. It uses linear field random field models incorporating non-local dependencies for information extraction, as presented in Finkel et al. [35].

Web pages do not always follow language formation standards, such as English or Portuguese, with several special symbols such as images, emojis, abbreviations without explaining their meaning, and many others. Thus, it is a demand to pre-process them. Next, we present the vastly used pre-processing steps.

**Tokenization** segments texts by checking the limits of characters, words, and punctuation. The components that result from segmentation are identified as *tokens*. There are special cases in which segmentation is not trivial. For example, the hyphen is used to separate syllables from words at the end of a line. In this way, when tokenizing, the two parts on different lines need to be reconstructed as a single word. Because of these special needs, ML models with labeled bases started to perform segmentation with accuracy reaching 99% [36].**Stemming** reduces words to their radicals by eliminating morphological variations and inflections. This process helps to normalize the text and reduce ambiguities. For example, in the English language, “Mr.”,“Mr”, “mister”, and “Mister” have the same meaning and must be represented in a single way [36].**Stopwords** are words that do not add meaning to the text, like prepositions and articles. Removing these words is a common strategy to retain more significant meaning to the sentences when retrieving information or computing frequencies.

We presented the main NLP concepts used to extract attributes from the databases used in popularity prediction studies. Thus, we present in a concise way the tasks of NER, sentiment analysis, text normalization to pre-process it, the LDA algorithm, and the models *n*-*gram*.

## 3. Popularity Prediction

Tatar et al. [8] summarized the studies on popularity prediction reporting the initial approaches that tried to understand the pattern of distribution of users’ access to different types of content [37]. After mapping these popularity probability distributions (mainly from videos), a search for models capable of predicting the popularity of information on the Web began [8]. It is interesting to note that researchers could use the popularity prediction in conjunction with other technologies such as: increasing the hit rate of cache relocation algorithms [12], optimization of news articles [10], associated with systems recommendation [38] among others. One of the first works that present a predictive method with ML was [22], showing a linear correlation between the different moments of the content life cycle on the Web. The predictive model presented was used in other studies, including the proposal of on-demand video service cache reallocation [12].

Our goal is to research and apply the ML methods used to predict the popularity of content on the web. To determine the search string for scientific articles, we use the following keywords: web content, popularity, prediction, classification, regression, feature selection, deep learning, word embedding, SVM, KNN, and Naive Bayes. The combinations of those keywords originate the following search string:
(“**web content**” OR “**popularity prediction**”) AND (“**machine learning**” OR “**classification**” OR “**regression**” OR “**feature selection**”) AND (“**deep learning**” OR “**word embedding**” OR “**SVM**” OR “**KNN**” OR “**Naive Bayes**”).

The databases selected for the research were Scopus, IEEE Xplore, and ACM Digital Library. We used as exclusion criteria papers that were not related to the prediction of popularity on the web, articles in other idioms that were not in English, and short texts with less than three pages. Using the search string, a search was performed on the IEEE Xplore database, returning 398 texts. From those, 387 were rejected as they did not fit the popularity prediction theme, leaving us with 11 articles. In the Scopus database, we obtained 573 papers with 547 excluded due to the three exclusion criteria, and, in the ACM Digital Library of 606 articles, 576 were discarded. All articles selected from the three bases added up to a total of 67 papers to be studied. We analyze and choose the most applied methods that will be explained in this manuscript.

This section presents the taxonomy built from the methods involved in Popularity Prediction. We present definitions, the operation of popularity predictions, the types of content, and a taxonomy to classify the models studied.

### 3.1. Taxonomy

To structure the study and presentation, we divided the methods of predicting popularity according to the problem definition and the prediction task, as follows:
**Regression Methods**. These methods perform a numerical prediction, quantifying the popularity according to the defined metric. The most common target attributes are the number of views, number of shares, number of tweets, and comments. These predictive methods use **Regression** and are often called regressors [9,22].**Classification Methods**. Popularity classes are defined; the predictive model allocates the content in one of the defined classes. The goal is to predict whether content will become popular or not; in most cases, only two classes are used: popular and non-popular. These predictive methods use **Classification** and are often called classifiers [13,15,16].

In addition to the above division, we can group the prediction methods according to the attributes used:
**Textual Attributes**. These attributes are extracted from the content using NLP techniques. The extraction can be direct from the content. In news articles, it can be from the description presented on the Web, as in videos and images, and even taking advantage of social media elements, such as comments published by users.**Visual Attributes**. These attributes are extracted from videos and images using ML techniques (ANN, for example) or manually selecting features from the frames representative of the content.**Metadata Attributes**. These attributes are provided by the website where the content was published and inherent to the Web. However, they do not belong to any previous groups, such as the source of the content, category, number of views, and publication date.

This taxonomy is shown in Figure 1:

Here, we classify the studies that present popularity prediction models using the proposed taxonomy. We show the results in Table 1, where *C* in predictive tasks indicates that the model studied uses Classification and *R* indicates a predictor that uses Regression. These surveys were also classified according to the attributes used. This classification is not exclusive. Some models are positioned in more than one category. These articles present several techniques and models that can be considered state of the art for predicting popularity. Table 2 shows the best models for each study and the performance earned. It is essential to pay attention to the metrics used to validate the comparisons. We observed that the classifiers that use textual attributes usually achieved the best results. In contrast, the regressors with the best results used visual features. The researchers can take this trend into account for future work.

### 3.2. Popularity Measure

Popularity of content is the relationship between an individual item and the users who consume it. Popularity is represented by a metric that defines the number of users attracted by the content being studied, reflecting the online community’s interest in this item [8]. Looking at the “most popular” videos or texts on the Web, the concept of popularity is intuitively understood. However, it is necessary to define objective metrics to compare two items and define which one is the most popular. Several measures point out which content attracts the most attention on the Web; that is, the number of users willing to consume the item searched. The “classic” web metric is the number of views. It is not always available, and, in some cases, does not represent the relationship of interest between the content and the users. Yao and Sun [41] revealed that the most viewed news articles are not always the most commented on and vice versa. This inference extends to the most shared items. In summary, defining the metric to be used, which may vary according to the context, is essential in a study on popularity prediction. In the literature, the main metrics and their respective meanings are as follows:
number of views, reflecting the number of users [11,22];number of shares, reflecting the notoriety of the content [10,16], andnumber of tweets and comments, reflecting the time that users spend on the content [13].

### 3.3. Content Types

Web Content is defined as an individual item available on a website in text, audio, image, or video format [8]. The attention of users on the Web is spread over several sites and various types of content. Some of the most popular are: videos produced by users, responsible for much of the Internet traffic; news articles shared and consumed on mobile devices; stories published in news aggregators; and items (comments, photos, videos) published on social networks [8]. The concepts presented can be applied to any type of content available online. However, to define the work scope, we will present methods and techniques that predict the popularity of videos, news articles using Machine Learning and Natural Language Processing.

**Videos Online**. YouTube [42], the largest online video platform uploading over 500 h per minute [43] and over one billion videos viewed per day, has been the main focus of previous work [8]. Studying the popularity of YouTube content is challenging due to the growing number of videos, the various features provided by the platform, and the limitations associated with selecting a representative subset of videos for the problem in question [44]. The number of views usually expresses the popularity of videos on the Web. It follows a long-tail distribution but depending on the set of videos chosen. A more detailed analysis reveals that different video’s activities follow similar patterns during periods of peak popularity [37].

**News Articles**. The primary source of information in the digital world, news articles, are distributed massively through social networks. While videos attract a user’s attention over a long period, interest in the news is temporary, with their attention span a few days after publication. The popularity metric often used is number of comments, as news platforms rarely disclose the number of views [8].

As each type of content has different characteristics, it is necessary to select the attributes that describe the content and its associated variables. Such a selection is known as feature engineering and is an essential part of the popularity prediction. The choice of attributes directly influences the quality of the predictive models. For this reason, several studies try to find a correlation between them and the final popularity of the content [45]. However, several factors that may also influence the popularity are difficult to measure, such as content quality, the relevance of the author, and users’ relevance.

There are some apparent attributes to select and others, not so obvious that strongly impact predictive models. Some influencing factors are already well established in the literature. For example, videos that evoke strong and positive emotions are among the most shared, in addition to being the ones that spread the most quickly [46]. Thus, conducting sentiment analysis to determine the content’s polarity results in an essential predictive attribute [10]. On the other hand, the definition of other attributes that make items popular could be hard. However, we have known that high-quality content is among the most viewed. Nevertheless, quality is a complex metric to measure. It involves subjective factors, making it challenging to capture attributes that represent the quality of the content. Another factor, not trivial to include in the predictive models, is the real world’s events that directly influence which virtual content will be most sought after, impacting its popularity. This has been a trend in items that go viral on the Internet [37].

Table 3 shows some of the most used predictive attributes: characteristics of the content creators, for example, the authors with the highest audience tend to have popular content just for their identity [13]; sentiment analysis and keywords that strongly impact popularity, both positively and negatively. In most studies, the categorization of content contributed positively to the prediction of popularity. Finally, attributes related to social networks such as the number of followers, online reputation, previous content that had many views, and a large number of shares also contribute to the increase in popularity [10].

### 3.4. Operation

For the realization of the popularity prediction, we used the traditional ML flow. Although the studies do not explicitly report, we will use the model presented by Khan et al. [16] as a guide for our work, which has the following steps:
Data collection. Obtaining the necessary data is not always a trivial task. Several studies use the YouTube and Twitter [47] platforms, which, despite not making the data openly available, there are several APIs facilitating this collection. Both platforms have a defined metric and an ecosystem that allows for determining the popularity of the published items.Pre-processing. In this step, we try to adjust the data for the use of the algorithms. In addition to searching for missing elements, we defined popularity classes to try to balance the data set.Feature selection. This step is the center of current popularity prediction research. The most recent work tries to automate this selection to try to get better results.Model Training. The prediction itself uses the models most relevant to the context of the problem to allow an efficient comparison with the research already carried out.Validation. In this stage, the performance of the models is tested and evaluated.

## 4. Classification Models

In this section, we present the ML models that use classifiers to predict Web content’s popularity. The definition of the class *popular* differs from one study to another, taking into account the dataset used and the context of the research. Usually, popular content belongs to the minority class, causing the popularity classes to become unbalanced, allowing the model to obtain high degree of correctness, without necessarily classifying them correctly. For this reason, several studies opt for employing a balancing strategy.

Performance evaluation and comparison between models is vital to find the best answer to the problem in question. Among the metrics used to evaluate the classifiers is **Accuracy**, defined by Equation (Equation 2). This metric is the complement of **Error Rate**, or incorrect classifications, presented in Equation (Equation 3). $\hat{f}$ is the classifier, $y_{i}$ the known class of $x_{i}$ and $\hat{f}\left(x_{i}\right)$ the predicted class, $\delta (y_{i},\hat{f}\left(x_{i}\right)=1$ if $y_{i}\neq \hat{f}\left(x_{i}\right)$ is true and 0, otherwise. Using as an example a problem of two classes, where one is popular content and the other unpopular, it is possible to present the Error Rate in a more understandable way as in Equation (Equation 4). FP are false positives, examples belonging to the unpopular class classified as popular and FN are false negatives, examples belonging to the popular class are classified as unpopular. [17]. As in the case of popularity prediction, popular content is in the minority. The algorithms that classify the content as unpopular tend to have better accuracy. In this context, it is worse to have many false negatives.
(2)$$ac\left(\hat{f}\right)=1-err\left(\hat{f}\right)$$
(3)$$err\left(\hat{f}\right)=\frac{1}{n}\sum_{1}^{n}\delta \left(y_{i},\hat{f}\left(x_{i}\right)\right)$$
(4)$$err\left(\hat{f}\right)=\frac{FP+FN}{n}$$

Still using the problem of two classes, we can mention other metrics used as the **precision**, defined by Equation (Equation 5), which presents the proportion of positive examples correctly classified among all those predicted as positive [17], **recall**, defined by Equation (Equation 6), which corresponds to the hit rate in the positive class. In Equations (Equation 5) and (Equation 6), TP is the number of true positives, FP are the false positives, and FN is the number of false negatives. These equations were defined for models of two classes [17]:
(5)$$prec\left(\hat{f}\right)=\frac{TP}{TP+FP},$$
(6)$$rev\left(\hat{f}\right)=\frac{TP}{TP+FN}.$$

The precision indicates the accuracy of the model, while the recall indicates completeness. Analyzing only the precision, it is not possible to know how many examples were not classified correctly. With the recall, it is not possible to find out how many examples were classified incorrectly. Thus, we usually performed with the F-measure, which is the weighted harmonic mean of precision and recall. In Equation (Equation 7), *w* is the weight that weighs the importance of precision and recall. With weight 1, the degree of importance is the same for both metrics. The measure $F_{1}$ is presented in Equation (Equation 8) [17]:
(7)$$F_{m}\left(\hat{f}\right)=\frac{\left(w+1\right)\times rev\left(\hat{f}\right)\times prec\left(\hat{f}\right)}{rev\left(\hat{f}\right)+w\times prec\left(\hat{f}\right)}$$
(8)$$F_{1}\left(\hat{f}\right)=\frac{2\times rev\left(\hat{f}\right)\times prec\left(\hat{f}\right)}{rev\left(\hat{f}\right)+prec\left(\hat{f}\right)}$$

The Receiving Operating Characteristics (ROC) graph [48] is represented in two dimensions with the *x*- and *y*-axis representing the measures of false positive rate (FPR) and true positive rate (TPR), respectively [17]. In this graph, the diagonal represents a random classifier, so the best models can classify above this line, as shown in Figure 2.

It is usual to construct a ROC curve to compare the performance between the different classification models, as seen in Figure 2, and calculate the area under ROC curve (AUC). For the construction of the ROC curve, it is necessary to order the test cases according to the continuous value provided by the classifier (depending on the model, an adaptation may be necessary) [48].

### 4.1. Textual Features

NLP techniques allow the extraction of several attributes directly from content, as in news articles, or from information provided, such as descriptions of videos and images. Among these techniques, there are the sentiment analysis, NER, subjectivity of the text, and discovery of topics with the LDA algorithm [31].

Twitter, one of the most popular social networks in the world, allows sharing information via short messages. News Articles are shared on Twitter by publishing the news URL and the retweet feature, which allows sending information without modification. Bandari et al. [13] used five classifiers with a set of multidimensional attributes to predict the popularity of news articles on Twitter through the number of *tweets* and *retweets*.

The news articles were collected from the news aggregator *Feedzilla* and the attributes which tried to cover different dimensions of the problem were:
The source of the news, which generated or published the article;The category of the article, according to *Feedzilla*;The subjectivity of the article’s language;Named entities present in the articles.

They collected data from 8 August 2011, to 16 August 2011, totaling 44,000 articles. For each article, the Topsy [49] tool provided the number of *tweets*. For the recognition of named entities (places, people, or organizations) the Stanford-NER tool was used. For the articles’ subjectivity, a Ling–Pipe classifier was used, which is a set of tools for NLP with ML algorithms developed in Pang and Lee [50]. To highlight the contribution of subjectivity in the analysis carried out, the authors sought two *corpus*: the first had a more informal language while the second was more objective and with less sentiment in the communication carried out. Next, the tool was trained with the transcripts of Rush Limbaugh [51] and Keith Olberman [52] as the subjective *corpus*. For the training of objective language, the transcripts of the CSPAN [53] were used, as well as the transcription of articles from the website *FirstMonday* [54].

The data collected showed that the absolute values sometimes did not represent the desired information. For example, the amount of news published in the health category does not emphasize its importance. This category has few published articles, but they are among the most shared. Thus, if we look only at the number of shares, a category with many articles would seem more important, even if your articles are not as shared. Due to this possibility of misinterpretation, the authors proposed the measure *t*-*density* (Equation (Equation 9)). Thus, a *t*-*density* was calculated for each category and for each source of the article [13]:
(9)$$t-density=\frac{Number\;of\;Tweets}{Number\;of\;links}$$

The dataset was divided into three classes, covering different ranges (*tweets* and *retweets* were counted as *tweets*): Class A with up to 20 *tweets*, Class B ranged from 20 to 100, and Class C with more than 100 *tweets*. Articles that were not shared on Twitter, that is, with 0 *tweets*, were not considered for the popularity prediction. Classifiers are induced from four ML methods: Bagging, Decision Tree J48, SVM, and Naive Bayes. They compare the performance of these models using accuracy as a metric. The results indicate that it is possible to predict the popularity *before* the publication with an accuracy of approximately 84% using a set of attributes extracted directly from the news articles’ content, with algorithm Bagging. Moreover, the classifiers trained to predict whether or not an article will be shared from Twitter using the same set of attributes have reached an accuracy of 66% [13].

Finding relevant textual attributes allows optimizing the content, in addition to prediction. In this sense, Fernandes et al. [10] propose an Intelligent Decision Support System (IDSS) to predict if a news article is popular or not and subsequently suggest simple changes in the content that would increase its popularity. The prediction module uses as inputs the digital media content (images, videos), previous popularity of the news referenced in the article, the average number of shares of keywords, and NLP attributes [10]. The news articles are gathered from the website Mashable [55] covering the period of two years. The metric for measuring popularity was the number of shares, and they considered a binary classification (popular/unpopular). To obtain a balanced distribution, the authors used the median number of shares, so articles with more than 1400 shares were considered popular.

The predictive results are explored in several ways. First, the five most relevant topics are identified in all the articles with the LDA algorithm [31]. After that, they measured the distance between each article and these topics. The results are incorporated as predictive attributes [10]. Regarding the subjectivity and polarity of the sentiment analysis, the authors adapted and used the Pattern module [56] developed in Smedt and Lucas [57]. Several attributes were extracted from the subjectivity and sentiment analysis, including subjectivity of the title, subjectivity of the text, polarity of the title, rate of positive and negative words, the polarity of the text, polarity of words, and rate of positive words between those that are not neutral and the rate of negative words among those that are not neutral.

The authors tested five classification methods: Random Forest (RF); Adaptive Boosting (AdaBoost), SVM with a Radial Base Function (RBF), KNN, and Naive Bayes (NB). The following metrics were computed: Accuracy, Precision, Recall, $F_{1}$ Score, and the AUC. The Random Forest has the best results with 0.67 of Accuracy and 0.73 of AUC.

From the results, they identified that, among the 47 attributes used, those related to keywords, proximity to LDA topics, and article category are among the most important. The optimization module seeks the best combination over a subset of features suggesting changes, for example, by changing the number of words in the title. Realize that it is the responsibility of the author of the article to replace the word. Applying the optimization to 1000 articles, the proposed IDSS achieved, on average, a 15% increase in popularity. The authors observed that NLP techniques to extract attributes from the content proved to be successful.

After the study was carried out in [10], the database was made available in the UCI Machine Learning repository allowing for new research and experiments. In 2018, Khan et al. [16] presented a new methodology to improve the results presented in [10]. The first analysis was to reduce features to two dimensions using Principal Component Analysis (PCA). PCA is a statistical procedure that uses orthogonal transformations to convert a set of correlated attributes into a set of linearly uncorrelated values called principal components. Thus, the two-dimensional PCA analysis output would be two linearly separated sets, but the results of that dataset did not allow this separation. Three-dimensional PCA analysis was applied to attempt linear separation, but it was also unsuccessful [16].

Based on the observation that the features could not be linearly separated and on the trend observed in other studies, the authors sought to test models of nonlinear classifiers and ensemble methods such as Random Forest, Gradient Boosting, AdaBoost, and Bagging. In addition to those, other models were tested to prove the effectiveness of the hypothesis like Naive Bayes, Perceptron, Gradient Descent, and Decision Tree. In addition, Recursive Attribute Elimination (RFE) was applied to obtain the 30 main attributes for the classification models. RFE recursively removes the attributes one by one, building a model with the remaining attributes. It continues until a sharp drop in model accuracy is found [16]. The classification task adopted two classes: popular articles with more than 3395 shares, and non-popular. Eleven classification algorithms were applied, showing that the ensemble methods obtained the best results, with Gradient Boosting having the best average accuracy. Gradient Boosting is a set of models that trains several “weak” models and combines them into a “strong” model using the gradient optimization. Gradient Boosting reached an accuracy of 79%, improving the result found in Fernandes et al. [10]. Other models have obtained interesting results as well; for example, the Naive Bayes model was the fastest, but it did not perform well because the attributes are not independent. The Perceptron model had its performance deteriorated as the training data increased, which can be explained by the data’s nonlinearity. Hence, the performance of the MLP classifier significantly improved the accuracy of the predictive task.

An exciting approach focusing on the attributes is presented in [15]. The authors hypothesized that the title’s grammatical construction and the abstract could emerge curiosity and attract readers’ attention. A new attribute, called *Gramatical Score*, was proposed to reflect the title’s ability to attract users’ attention. To segment and markup words, they relied on the open-source tool Jieba [58]. The *Grammatical Score* is computed followed the steps below:Each sentence was divided into words separated by spaces;Each word received a grammatical label;The quantity of each word was counted in all items;Finally, a table with words, labels, and the number of words was obtained;Each item receives a score with the Equation (Equation 10), where $gc_{i}$ represents the Grammatical Score of the ith item in the dataset and *k* represents the kth word in the ith item. The *n* is the number of words in the title or summary. The weight is the amount of the kth word in all news articles, and count in this equation is the amount of the kth word in the ith item:
(10)$$gc_{i}=\sum_{k=0}^{n}weight\left(k\right)\times count\left(k\right)$$

In addition to this attribute, the authors used a logarithmic transformation and normalization by building two new attributes: *categoryscore* and *authorscore*:
(11)$$categoryscore=\frac{\sum_{n}ln\left(s_{c}\right)}{n}$$

The *categoryscore* is the average view for each category. The variable *n* in the Equation (Equation 11) represents the total number of news articles of each author. For each category, the data that belonged to this category were selected, and Equation (Equation 11) was used:
(12)$$authorscore=\frac{\sum_{m}ln\left(s_{a}\right)}{m}$$

The *authorscore* is defined in Equation (Equation 12), where *m* represents the total number of news articles of each author. Before calculating the *authorscore*, data are grouped by author. For the prediction, the authors used the titles and abstracts’ length and temporal attributes in addition to the three mentioned attributes. The authors’ objective was to predict whether a news article would be popular or not. For this, they used the freebuf [59] website as a data source. They collected the items from 2012 to 2016, and two classes were defined: popular and unpopular. As these classes are unbalanced and popular articles are the minority, the metric AUC was used, which is less influenced by the distribution of unbalanced classes. In addition, the kappa coefficient was used, which is a statistical measure of agreement for nominal scales [60]. The authors selected five ranking algorithms to observe the best algorithm for predicting the popularity of news articles: Random Forest, Decision Tree J48, ADTree, Naive Bayes, and Bayes Net. We identified that the ADTree algorithm has the best performance with 0.837 AUC, and the kappa coefficient equals 0.523.

Jeon et al. [40] proposed a hybrid model for popularity prediction and applied it to a real video dataset from a Korean Streaming service. The proposed model divides videos into two categories, the first category, called A, consisting of videos that have previously had related work, for example, television series and weekly TV programs. The second category, called B, is videos that are unrelated to previous videos, as in the case of movies. The model uses different characteristics for each type. For type A, the authors use structured data from previous contents, including the number of views. For type B, they use unstructured data such as texts from titles and keywords.

The XGBoosting algorithm, a model developed for rapid development and classification based on parallel processing, was used to predict a type A video. The authors use ANN with embedding techniques to obtain generation prediction resources for type B videos. They used Continuous Bag-of-Words (CBOW) through Word2Vec to build embeddings. In the end, they concatenate predictions of both models to deliver the final result. In addition to title and keywords, they use actor names, television channel names, and episode counts for feature extraction. The use of embeddings to obtain the title characteristics improved the prediction performance compared to the other four models with the same dataset [40].

### 4.2. Visual Features

Most studies use the textual attributes and meta-attributes provided by the sites. However, in recent years, with technological advances, it has become possible to also use visual attributes extracted directly from videos. One of the first studies in this regard was [11]. The authors studied the problem of predicting the popularity of videos shared on social networks. The prediction was treated as a classification task, and the attributes were extracted directly from the videos using a Deep Neural Network (DNN) architecture. The authors postulated that, if the predictive model incorporated the sequential information presented in the videos, a better classification accuracy would be obtained. The DNN is a Long Term Recurrent Convolutional Network (LRCN) [61] that is able to take into account the order of the information when learning the weights. They called this method *Popularity*-*LRCN* and evaluated it with a dataset of 37,000 videos collected from Facebook [62].

The network architecture is composed of an input layer that supports 18 frames of 227 × 227 × 3 dimension for each video. There are other eight layers, where the first five are convolutional layers, the sixth layer is a completely connected layer with 4096 neurons, the seventh is a Long Short-Term Memory (LSTM), and the last layer is the classification layer with two neurons. They used *softmax* in the classification layer [11]. To increase the network invariance, layers of *max pooling* were used after the first, second, and fifth convolutional layers. ReLU was used as a nonlinear activation function applied to all convolutional layers’ outputs and the layers completely connected. During the training, the 320 × 240 × 3 video frames were randomly reduced to 227 × 227 × 3. In addition, a mirroring technique was used to increase the amount of sample in the training dataset. The network has been trained over 12 epochs with 30,000 iterations each [11].

Data were collected from videos shared on Facebook from 1 June 2016 to 31 September 2016. Due to the massive difference in the videos’ number of views (videos with millions of views and videos watched less than 1000 times), authors used a logarithmic transformation. In addition, in order to reduce the bias introduced by the fact that content producers with a large number of followers attract a large number of views, the authors included in the standardization procedure the number of followers of producers [11]. Thus, the normalized popularity score (*NPS*) is calculated using Equation (Equation 13):
(13)$$NPS=\log_{2}\left(\frac{viewcount+1}{number\;of\;publisher^{\prime }s\;followers}\right)$$

After normalization, the dataset was divided into two classes: popular and non-popular. The normalized popularity median enables a balanced distribution of classes. The authors compared the proposed method with two traditional classifiers, namely, a logistic regression, and SVM with a radial base function. The input attributes to the classifiers are as follows:
HOG: a resource descriptor with 8100 dimensions called Histogram of Oriented Gradients [63];GIST: a resource descriptor with 960 dimensions [64];CaffeNet [65]: a vector of features with 1000 dimensions extracted from a convolutional network;ResNet: a vector of features with 1000 dimensions extracted from DNN [66].

The *Popularity*-*LRCN* surpassed the other methods reaching 70% accuracy. The way that came closest was the SVM when used with features from CaffeNet or ResNet, which had an accuracy of approximately 65%. Features from HOG and GIST had less influence on the final result, reaching a maximum accuracy of 60% with logistic regression and SVM and emphasizing that Popularity-LRCN used raw frames without performing attribute engineering.

We present the main popularity classification methods found in the literature. We observed that most of them rely on the extraction of attributes through NLP techniques, even when analyzing videos’ popularity. Recently, Trzcinski et al. [11] presented a model that extracts features directly from the videos, expanding the search horizon.

## 5. Regression Models

In this section, we present the methods used to perform the numerical popularity prediction. The goal is to quantify the degree of popularity. The applications for this type of prediction are diverse, among them, the proactive allocation of resources. It is usual to indicate the prediction error, the correlation coefficient (Pearson or Spearman), and the determination coefficient ($R^{2}$).

The determination coefficient indicates the degree to which the prediction made exceeds the average of the desired value. Thus, the value of $R^{2}$, which can vary from 0 to 1 calculated according to Equation (Equation 14), indicates the percentage of variation related to predictive features. ${\hat{N}}_{c}\left(t_{i},t_{r}\right)$ is the popularity prediction of the item *c* for the instant $t_{r}$ realized at the instant $t_{i}$ and $N_{c}\left(t_{r}\right)$ is the actual popularity at time $t_{r}$. For example, a value of 0.81 indicates that 81% of the variation in popularity can be explained based on the predictive variables used. In comparison, the other 19% are not related to the chosen features. Thus, this metric indicates the quality of the numerical predictions made [67]:
(14)$$R^{2}=1-\frac{\left[\sum {\left(N_{c}\left(t_{r}\right)-{\hat{N}}_{c}\left(t_{i},t_{r}\right)\right)}^{2}\right]/\left(n-2\right)}{\left[\sum {\left(N_{c}\left(t_{r}\right)-{\bar{N}}_{c}\left(t_{i},t_{r}\right)\right)}^{2}\right]/\left(n-2\right)}$$

Another metric widely used to evaluate regressors is the correlation between the predicted popularity and the examples’ verified popularity. The correlation determines the strength of the relationship between these two variables. It is important to note that the correlation does not indicate causality, but it does denote the quality of the prediction. Pearson’s correlation coefficient measures the linear dependence between the predicted values and the actual popularity. The values of the correlation coefficient can vary between –1 and 1. They will be positive when high values of the prediction correspond to high values of popularity and negative when the prediction’s high values correspond to low values of popularity. We want to find values as close to 1 as possible [67].

Spearman’s correlation coefficient is a well-known metric for rankings assessments. Like a Pearson’s coefficient, the values are in the range of −1 to 1, and the positive value indicates agreement while the negative values indicate disagreement. Compared to Pearson’s coefficient, it is easier to calculate and less susceptible to outlier values [67].

Another way to evaluate the regression models is by calculating the prediction errors. The Least Squares Error consists of an estimator that minimizes the sum of the regression residuals’ squares to maximize the degree of adjustment of the model to the observed data as we can see in Equation (Equation 15). The least squares method requirement is that the unpredictable factor (error) is randomly distributed, and this distribution is normal. The Relative Square Error (RSE) would express the error if an average predictor were used. Thus, the RSE takes the total squared error and normalizes it by dividing it by the simple predictor’s absolute squared error like in Equation (Equation 16):
(15)$$LSE=\sum_{c}{\left[{\hat{N}}_{c}\left(t_{i},t_{r}\right)-N_{c}\left(t_{r}\right)\right]}^{2}$$
(16)$$RSE=\sum_{c}{\left[\frac{{\hat{N}}_{c}\left(t_{i},t_{r}\right)-N_{c}\left(t_{r}\right)}{N_{c}\left(t_{r}\right)}\right]}^{2}$$

### 5.1. Textual Features

Oghina et al. [14] demonstrated that it is possible to predict IMDB’s [68] film scores using multiple social networks. Modern Information retrievers use various sources of information to achieve their purpose, called the Cross Channel Prediction Task. That is, information from different sources (websites) are analyzed to make predictions on another channel [14]. The social networks chosen were Twitter and Youtube, obtaining the following quantitative attributes: number of views, number of comments, number of favorites, number of likes, number of dislikes, the fraction of likes over dislikes for each YouTube video, and number of tweets on Twitter. The value of each attribute is the natural logarithm of its frequency. In addition to those attributes, textual features were extracted by comparing the log-likelihood function of a term in two corpora to identify the words and phrases indicative of positive and negative moods like [69]. These corpora are tweets about the analyzed films and comments about the movie trailers on YouTube. Examples of extracted positive textual features include the stems amaz, awesom; negative ones include worst, terribl. Thus, the frequency of these words could be used as features.

The dataset consisted of 70 films, with the notes reported on 4 April 2011. Ten films were kept separate for extracting textual attributes of this set, leaving 60 movies for testing. The dataset was supplemented with data from Twitter, 1.6 M tweets published between 4 March 2011, 4 April 2011, and 55 K YouTube comments. The authors used linear regression with WEKA implementation comparing the experiments using Spearman’s coefficient (ρ) [14]. The baseline of the experiments was the prediction made only with quantitative data. Next, the authors included the textual attributes of Twitter, and later the textual attributes of YouTube were included in the predictive models. The performance, including the textual characteristics of YouTube data, worsened the result while the model with the Textual features of Twitter obtained a performance superior to baseline. Combining the Textual attributes of Twitter with those of YouTube, the baseline was surpassed, but the performance continued to be inferior to the Twitter model.

Evaluating the correlation of quantitative attributes with the prediction result, the authors discovered that the fraction of likes about dislikes is the best predictive attribute. This quantitative attribute was tested with the textual features, obtaining the best predictive result with a Spearman’s coefficient (ρ) of 0.8539 [14]. In addition to comparing the Classifiers, Bandari et al. [13] (presented in Section 4.1) used the same attributes with three regressors: linear regression, KNN, and SVM. The attempt was to predict the exact number of tweets an article would receive. The best result found using the determination coefficient ($R^{2}$) as a comparison metric, with linear regression, was 0.34. With this performance, we cannot say that these models are good enough to predict the exact amount of tweets an article will receive.

Liu et al. [15] made another unsuccessful attempt to use regression with textual attributes. Using the same features presented in Section 4.1, the WEKA linear regression, and the determination coefficient ($R^{2}$) as a metric, the authors obtained unsatisfactory results. They attempt to use the *Grammatical Score* feature to improve the results, achieving a 6.62% increase in performance, obtaining a final result of the determination coefficient ($R^{2}$) of 0.5332.

### 5.2. Meta-Data Features

Although we present several methods that use different predictive attributes, it is possible to perform a popularity prediction using only the number of online content views. However, it can only be employed after the content is published, by capturing the number of views in an instant $t_{i}$ to predict the popularity in the instant $t_{r}$, with $t_{i}<t_{r}$. This simple idea brought good results when the dataset is from two sharing portals, namely, Digg [70], a news portal, and Youtube [22]. With Digg news, it is possible to predict the 30th day’s popularity using the number of views obtained in the first two hours. For Youtube, it is necessary to use the views obtained during the first ten days to predict the popularity on the 30th day. The explanation is the fact that the life cycles on both types of shared contents are different [22].

The news has a short life cycle, with a quick peak of popularity, but the interest is dispersed at the same speed. Videos have a continually evolving growth rate and, as a consequence, a longer life cycle. The likelihood of a video attracting much attention on the Web, even after its peak of popularity, is greater than the news articles [22]. Szabo and Huberman [22] found a strong correlation (Pearson’s coefficient above 0.9) between the logarithmic popularity in two distinct moments: the content that receives many views at the beginning tends to have a greater number of views in the future. The correlation found is described by a linear model with Equation (Equation 17):
(17)$$ln\;N_{c}\left(t_{2}\right)=ln\;r\left(t_{1},t_{2}\right)+ln\;N_{c}\left(t_{1}\right)+\varepsilon_{c}\left(t_{1},t_{2}\right)$$

$N_{c}\left(t\right)$ is the popularity of the item *c* from publication to time *t* and $t_{1}$ and $t_{2}$ are two arbitrarily chosen moments, with $t_{2}>t_{1}$. $r(t_{1},t_{2})$ is the linear relationship found between the logarithmic popularity and is independent of *c*. $\varepsilon_{c}$ is the noise term that describes the randomness observed in the data [22]. Szabo and Huberman [22] present three predictive models with error functions to be minimized using regression analysis. The first model uses linear regression applied on a logarithmic scale, the function to be minimized is the estimated least squares error (LSE) presented in Equation (Equation 15). ${\hat{N}}_{c}\left(t_{i},t_{r}\right)$ is the popularity prediction of the item *c* for the instant $t_{r}$ realized at the instant $t_{i}$ and $N_{c}\left(t_{r}\right)$ is the actual popularity at time $t_{r}$.

The regression model that minimizes this function is presented in Equation (Equation 18). In this equation, $\beta_{0}\left(t_{i}\right)$ is the regression coefficient and $\tau_{0}^{2}$ is the residual variation on the logarithmic scale:(18)$${\hat{N}}_{c}\left(t_{i},t_{r}\right)=exp\left[ln\;N_{c}\left(t_{i}\right)+\beta_{0}\left(t_{i}\right)+\tau_{0}^{2}/2\right]$$

The second model assumes that the evolution of popularity obeys a constant scale of growth. The error function to be minimized is the relative quadratic error (RSE) and is presented in Equation (Equation 16).The linear correspondence found between the popularity rates in early times and future times suggests that the expected popularity value, ${\hat{N}}_{c}\left(t_{i},t_{r}\right)$, for item *c* can be expressed as:(19)$${\hat{N}}_{c}\left(t_{i},t_{r}\right)=\alpha \left(t_{i},t_{r}\right)N_{c}\left(t_{i}\right)$$

$\alpha (t_{i},t_{r})$ is independent of the item *c*, but will depend directly on the error function you want to minimize. In this specific case, to minimize RSE, we will have:
(20)$$\alpha \left(t_{i},t_{r}\right)=\frac{\sum_{c}\frac{N_{c}\left(t_{i}\right)}{N_{c}\left(t_{r}\right)}}{\sum_{c}{\left[\frac{N_{c}\left(t_{i}\right)}{N_{c}\left(t_{r}\right)}\right]}^{2}}$$

The average growth profile of the training set’s popularity is the base of the third predictive model. The average of the submissions’ popularity at the time $t_{i}$ normalized by the popularity at the time $t_{r}$ represents growth profile:
(21)$$P\left(t_{i},t_{r}\right)={\langle \frac{N_{c}\left(t_{i}\right)}{N_{c}\left(t_{r}\right)}\rangle }_{c}$$

In Equation (Equation 21), ${\langle .\rangle }_{c}$ is the average of the standardized popularity over the entire training set. The prediction for an item *c* is calculated with the Equation (Equation 22):
(22)$${\hat{N}}_{c}\left(t_{r}\right)=\frac{N_{c}\left(t_{i}\right)}{P(t_{i},t_{r})}$$

The models presented by Szabo and Huberman [22] are simple and efficient. Their results indicate that it is possible to predict future popularity based only on the number of initial views, but they have some flaws. The models use the total number of views until $t_{i}$ as input, but two items can have similar number of views in $t_{i}$ and very different numbers of popularity rates in $t_{r}$. Thus, Pinto et al. [23] present two predictive models that try to correct these flaws and surpass the models presented in [22]. Instead of using the total number of views obtained in $t_{i}$, these views are divided into regular measurement intervals from publication to the time $t_{i}$, each interval is called delta popularity. Pinto et al. [23] proposes a Linear Multivariate (MLM) model that predicts popularity at instant $t_{r}$ as a linear function shown in Equation (Equation 23):
(23)$${\hat{N}}_{c}\left(t_{r}\right)=\Theta \left(t_{i},t_{r}\right)X_{c}\left(t_{i}\right)$$

Let $x_{i}\left(c\right)$ be the number of views received in the time interval $\delta_{i}$ and $X_{c}\left(t_{i}\right)$ the popularity vector for all ranges up to $t_{i}$, so we have the following representation: $X_{c}\left(t_{i}\right)={\left[x_{1}\left(c\right),x_{2}\left(c\right),x_{3}\left(c\right),\cdots ,x_{i}\left(c\right)\right]}^{T}$. The model parameters, $\Theta \left(t_{i},t_{r}\right)=\left[\theta_{1},\theta_{2},\cdots ,\theta_{i}\right]$ are computed to minimize the mean of the relative square error (MRSE), Equation (Equation 24):
(24)$$MRSE=\frac{1}{c}\sum_{c}{\left[\frac{{\hat{N}}_{c}\left(t_{i},t_{r}\right)-N_{c}\left(t_{r}\right)}{N_{c}\left(t_{r}\right)}\right]}^{2}$$

The idea is that, due to the different weights attributed to the time intervals observed in the history of the items, the MLM model can capture the pattern of evolution of the content’s popularity. However, this model is still limited, especially in videos that show different patterns of popularity evolution. A possible solution would be to create a specialized model for each known pattern, but the great difficulty is how to know, a priori, what will be the evolution pattern of the video to be predicted [23]. Thus, [23] chose to build a model that takes into account the similarity (number of views, up to $t_{r}$) between the video and known examples from the training set. This similarity is used to adapt to the popularity prediction. To measure the similarity between the videos, an RBF was used, which depends on the distances from the center’s entrance. Given the training set $v_{c}$, the parameter τ, the video *v*, and the MLM X(v) model, the authors created a Gaussian RBF presented in Equation (Equation 25):(25)$$RBF_{v_{c}}\left(v\right)=e^{\left(-\frac{||X\left(v\right)-X\left(v_{c}\right){\left|\right|}^{2}}{2.\tau^{2}}\right)}$$

In the experiment, a random sample of the training set was selected to be used as the center of Equation (Equation 25). For each video *v*, they computed $RBF_{v_{c}}\left(v\right)$, *C* is the sample used as the center, $w_{v_{c}}$ is the weight of the model associated with RBF *feature*, this model was called MRBF and is formally defined by the Equation (Equation 26) [23]:
(26)$$\hat{N}\left(v,t_{i},t_{r}\right)=\underset{model\;MLM}{\underbrace{\Theta_{\left(t_{i},t_{r}\right)}.X\left(v\right)}}+\underset{RBF\;features}{\underbrace{\sum_{v_{c}\in C}w_{v_{c}}.RBF_{v_{c}}\left(v\right)}}$$

Finally, in [23], the models are compared with the constant growth model of [22] called the S-H model, Equation (Equation 19). The models were compared by applying them to a YouTube video dataset, the error metric used was the MRSE, and the indication and reference times for the models were: $t_{i}=7$ and $t_{r}=$ 30. As expected, the MRBF Model obtains the best performance.

Hoiles et al. [39] presented a study with the purpose to analyze how metadata contribute to the popularity of videos on YouTube. The dataset was provided by BBTV and includes the metadata for the BBTV videos from April 2007 to May 2015 on YouTube. There were about 6 million videos distributed on 25,000 channels. By applying various ML algorithms to analyze the correlation of attributes provided by YouTube, the authors listed the five most important ones for increasing popularity: number of views on the first day of the video, number of subscribers to the channel, thumbnail contrast, Google hits (number of results found with the Google search engine when entering the video title), and number of keywords. The application of several ML algorithms to determine the number of views had the best result of the Conditional Inference Random Forest [71] with the determination coefficient ($R^{2}$) of 0.80 [39].

Another interesting finding was that the publication of videos outside the days scheduled for the videos’ launch tends to increase the number of views. In addition, the authors demonstrated that the optimization of the features allows the increase in popularity. As an example, we have that the title’s optimization increases the traffic due to the YouTube search engine [39]. The authors also presented a generalization of the Gompertz model presented in [72] to add external events, as shown in Equation (Equation 27). There ${\bar{v}}_{i}\left(t\right)$ is the total view count for video *i* at time *t*, u(.) is the unit step function, $t_{0}$ is the time the video was uploaded, $t_{k}$ with $k\in 1,\cdots ,K_{max}$ are the times associated with the $K_{max}$ exogenous events, and $w_{i}^{k}\left(t\right)$ are Gompertz models which account for the view count dynamics from uploading the video and from the exogenous events. In this way, they can identify the number of views from subscribers to the channel, non-subscribers, and increased views due to external events [39]:
(27)$$\begin{array}{c} {\bar{v}}_{i}\left(t\right)=\sum_{k=0}^{K_{max}}w_{i}^{k}\left(t\right)u\left(t-t_{k}\right),\\ w_{i}^{k}\left(t\right)=M_{k}\left(1-e^{-\eta k\left(e^{b_{k}\left(t-t_{k}\right)}-1\right)}\right)+c_{k}\left(t-t_{k}\right) \end{array}$$

### 5.3. Visual Features

Khosla et al. [21] were one of the first works to use visual information to predict the number of views that images would obtain on the Web. The data were extracted from the Flickr [73] site, as the authors wanted to use the image publishers’ social information. The attributes taken from the images were:
**Color histogram**: the authors used 50 colors as described in [74], marking each pixel of the image for those colors, creating a histogram of colors.**Gist**: a resource descriptor with 512 dimensions [64].**Texture**: they used the famous Local Binary Pattern (LBP) obtaining a descriptor with 1239 dimensions [75].**Color patches**: They used 50 colors as described in [74], in a bag-of-words representation obtaining a final vector with 4200 dimensions.**HOG**: a resource descriptor with 10,752 dimensions [63].**ImageNet**: They used deep learning to learn a representation of the image visible in a vector of 4096 dimensions [76].

The attributes are the input for a support vector regression (SVR) with a linear kernel to predict the number of views of an image, reaching a Spearman’s coefficient (ρ) of 0.40. The model was fed only with visual attributes. When using only the social attributes—number of friends or number of photos uploaded—of the image publisher with the same model, Spearman’s coefficient (ρ) was 0.77. The best result was defined by combining the visual and social features, reaching the mark of 0.81 [21]. Although this experiment demonstrates that the publisher’s social contacts have more results for the generation prediction than the images’ content, the visual attributes are essential to increase the prediction result. Other important factors were that colors closer to red tend to have more visualizations. In addition, the authors searched for the correlation of some objects produced in the images. A list of items was obtained, which, when obtained in the pictures, tend to have fewer views as examples: spatula, plunger, laptop, golf cart, space heater [21].

Trzcinski and Rokita [9] proposed a regression method to predict the popularity of online videos using SVM with Gaussian radial base function, called *Popularity*-*SVR*. This method, when compared to the models presented in [22,23], is more accurate and stable, possibly due to the nonlinear character of *Popularity*-*SVR*. In the comparison experiments, two sets of data were used, with almost 24,000 videos taken from YouTube and Facebook. This work also shows that the use of visual attributes, such as the output of DNN or scene dynamics metrics, can be useful for predicting popularity, even because they are obtained before publication. The accuracy of the prediction can be improved by combining initial distribution patterns, as in the models of [22,23], with visual and social attributes such as the number of faces that appear in the video and the number of comments received by the video. The visual attributes used were:
**Characteristics of the videos**. Simple characteristics were used, such as length of the video, the number of frames, resolution of the video, and the frames’ dimensions.**Color**. The authors grouped the colors into ten classes depending on their coordinates in the HSV representation (hue, saturation, value): black, white, blue, cyan, green, yellow, orange, red, magenta and others. The predominant color was discovered for each frame, classifying it in one of these ten classes.**Face**. Using a face detector was counted the number of faces per frame, the number of frames with faces, and the region’s size with faces in relation to the size of the frame.**Text**. Combining *Edge Detection* (image processing technique to determine points where light intensity changes suddenly) and morphological filters, regions of the video with printed text were identified, generating the following features: number of frames with printed text and the average size of the region with text in relation to the frame size.**Scene Dynamics**. Using the *Edge Change Ration* algorithm, the authors determined the limits of *shots* (series of consecutive images representing a continuous action). The number of *shots* and the average size of *shots*, in seconds, were used as attributes [77].**Clutter**. This measure represents the disorder of the video, the authors used the *Canny edge detector* to quantify the *clutter* [78]. The attribute used was the average of the detected pixels’ proportion and the number of pixels in a frame.**Rigidity**. To estimate the rigidity of the scene, the authors estimated the homography between two consecutive frames by combining the use of FAST [79], and BRIEF [80]. The attribute was the average of the number of valid homographs found.**Thumbnail**. The popularity for the *thumbnail* of the video was computed using the *Popularity API* following the work of [21].**Deep Features**. A 152-layer convolutional neural network called ResNet-152 [66] was used. For each video, a set of *thumbnails* per scene was extracted and propagated through ResNet-152. The output obtained was a vector of 1000 dimensions. This vector has been normalized resulting in a single value.

The predictive features include the visual attributes above and social characteristics such as the number of shares, *likes*, and comments. The predictive methods used for comparison are those presented in [22,23] and explained in Section 5.2. The MRBF regression model, explained by the Equation (Equation 26), presents the combination of two methods: the MLM regression model (linear) and RBF *features* (nonlinear). It is not necessary to perform this prediction in two stages. Inspired by the results of the MRBF, the *Popularity*-*SVR* uses a Gaussian RBF as the transformation kernel, allowing for mapping the vector of attributes in a nonlinear space where the relationships of the evolution patterns of the videos are easier to capture [9].

SVM with linear kernels create separation surfaces for linearly separable datasets or that have an approximately linear distribution. However, in nonlinear problems, this is not possible. This linear separation can be achieved by mapping the inputs from the original space to a larger space [17]. Let Φ:X→ℑ be a mapping, where *X* is the input space and *ℑ* denotes the feature space. The appropriate choice of Φ means that the training set mapped to *ℑ* can be separated by a linear SVM [17]. A kernel *K* is a function that receives two points $x_{i}$ and $x_{j}$ in the input space and calculates the scalar product of these objects in the characteristics space, mapping the input set in a new space dimensional [17].

As a result, the nonlinear characteristic of the transformation RBF kernel allows for a robust prediction based on similarity with the popularity evolution patterns identified in the training set. This proposal differs from the MRBF model that compares similarity with a set of videos selected at random from the training set [9]. The selection of the correct kernel can influence the performance of the model. For this reason, they search further for an optimal kernel. The popularity of a video *v* using the *Popularity*-*SVR* method can be calculated as in Equation (Equation 28) [9]:
(28)$$\hat{N}\left(v,t_{i},t_{r}\right)=\sum_{k=1}^{K}\alpha_{k}.\Phi \left(X\left(v,t_{i}\right),X\left(k,t_{i}\right)\right)+b$$

In Equation (Equation 28), $\Phi \left(x,y\right)=exp\left(-\frac{{\left||xy|\right|}^{2}}{\sigma^{2}}\right)$ is an RBF Gaussian parameter σ, $X(v,t_{i})$ is the vector of attributes for the video *v* available at time $t_{i}$ and ${X(k,t_{i})}_{k=1}^{K}$ is the set of support vectors returned by the SVR algorithm with the set of coefficients ${\alpha_{k}}_{k=1}^{K}$ and intercepts *b*. The authors found optimal values for the *C* hyperparameter of the Support Vector Machine optimization and σ for the RBF kernel using a search grid with the Python scikit’s *sklearn.grid*_*search.GridSearchCV* method in a preliminary set of experiments. The values found were: C=10 and σ=0.005 [9].

The *Popularity*-*SVR* was compared with other regression models using two sets of data. The first dataset was composed of YouTube videos, and the second dataset, also from videos, was extracted from different Facebook profiles. First, *Popularity*-*SVR* was compared with the prediction model presented in [22], which we will call the SH model, and the MLM and MRBF models presented in [23] using the number of views of YouTube videos with $t_{i}=6\;days$ and $t_{r}=30\;days$. The metric used for comparison was Spearman’s correlation coefficient.

The other comparison used the Facebook dataset, testing the models only with the number of views, then only with the social data, only with the visual attributes, and combining all of them. This last test was combining the social, visual attributes, and the number of views. Predicting with the visual information had the worst performance. However, when all the attributes are combined, the prediction is more accurate, proving the advantage of using all the sets of attributes in a combined way.

The *Popularity*-*SVR* method proposed in [9] is an evolution of the methods presented in [22,23], surpassing them in performance. In addition, the use of a set of visual attributes combined with the number of views and social data of the videos increases the popularity of the predictor’s performance. This information can be extracted from the videos before publication and can be used in other prediction models.

## 6. Case Study

After reviewing the literature, we identified that most previous research that have proposed methods for predicting the popularity of videos relying on textual attributes gather them from the title, but not from the videos’ content description. Among the works found in the literature, Fernandes et al. [10] is the one that engineers the most significant number of features to predict popularity. Thus, we use Fernandes et al. [10] as an inspiration for obtaining features not only from the title but also directly from the video descriptions in this work.

In this section, we present the case study methodology, which is composed of four phases divided as follows: (i) Data Collection, (ii) Extraction of features engineered from the textual content, (iii) Extraction of Word Embeddings, and (iv) Popularity Classification.

### 6.1. Video Communication

We can evaluate the user’s Quality of Experience (QoE) according to several metrics, among which we can highlight: initial playback delay, video streaming quality, quality change, and video rebuffering events. Loh et al. [81] developed ML models to estimate the playback behavior, it being possible to carry out monitoring that allows for adjusting the buffer size, improving the transmission quality. As it is impossible to monitor every packet of every video stream, service providers look for intelligent techniques and strategies to predict a change in quality in the transmission to adjust the necessary parameters and provide a better quality of user experience. We propose to obtain popular videos before they are published by extracting textual features from the video’s description. In this way, predictions and monitoring about the quality of streaming for the end-user can focus on the most significant videos, requiring a smaller quantity, improving performance for service providers and network operators who can better scale the necessary size of the buffer and improve QoE. In other words, our model can be used to identify videos that will demand more resources from the network infrastructure, allowing service providers to adopt preventive measures to maintain transmission quality.

New technologies to improve the efficiency of video transmission have attracted attention. Kim et al. [82] investigate how to improve the efficiency of video streaming using client cache. This work proposes a cache update scheme using reinforcement learning. The results demonstrate that the proposed cache update scheme reduces the amount of XOR operations in cache management, decreasing the number of transmissions by 24%. Again, identifying popular videos before publication allows reinforcement learning training to be utilized with a set of more meaningful videos, optimizing performance.

### 6.2. Data Collection

Our data are collected from Globoplay [83]. It uses the NGINX [84] software to manage HTTP requests [85,86]. This software records a log message for each video segment transmitted. We access the logs of requests from the live services and Globoplay’s on Demand Videos (VOD) [87,88]. We downloaded the records stored from 25 January 2021 to 1 March 2021. As the number of logs and videos is huge, we removed a sample space representing the total content. The goal is to use ML models to tell whether a video will be popular or not. For this, we extract from the logs (i.) the number of views, (ii.) the number of bytes transmitted for each video, (iii.) the URL, and (iv.) the code of the video. After this step, we enriched the data with title information and description of the videos retrieved from the Globoplay website with the BeautifulSoup [89] library so that we could extract textual features and embeddings from them.

The dataset consists of 9989 videos, distributed as movies, series, entertainment, and news categories. Thus, our set is quite heterogeneous, and there is no predominance of video genres that can influence the prediction results. The most viewed video has 75,754 views. As the logs do not automatically record this value, we had to calculate it from the HTTP requests. Thus, all accesses made by the same user to the same video during 30 min count as just one view. This calculation can decrease the number of total views, but it does not interfere with the analysis. Figure 3 shows the complementary cumulative distribution function of probability for the Globoplay videos visualization, presented in log scale. From the graphic, we realize that the curve presents a long-tail behavior, which means that most of the visualizations occur to a small fraction of videos. For instance, only 6% of videos have more than 1000 views, while 50% have less than 20 views. The quartiles of the set of videos were measured, with the third quartile equal to 83. That is, only 25% of the videos have more than 83 views. If we look at videos with more than 1000 views, we will see that they represent just over 6% of the total videos. We can see this information in Figure 3. Another interesting piece of data are the sum of the views of the videos: 6% of the most popular videos have 85% of the number of views as we can see in Figure 4. These same videos correspond to 73% of the payload carried in bytes. We can see this information in Figure 5.

In Table 4, we see that 616 videos with more than 1000 views correspond to 85% of our dataset’s total number of views. These data corroborate that few videos concentrate most of the users’ attention. Another important fact is that, by adding the videos between 83 and 1000 views (1875) and those with more than 1000 views (616), we get that 25% of our dataset is responsible for 93% of the total bytes transmitted. Thus, when forecasting videos with more than 83 views, we anticipate which videos will use more than 90% of the infrastructure of streaming services. For this reason, when defining the popularity class in our experiments, we will use the value of the third quartile.

### 6.3. Textual Features

To extract textual features, we used Fernandes et al. [10] as a guide. We tried to get as many similar features as they have as possible. However, due to the difference in information provided by the platforms (they used Mashable [55] while we use Globoplay), we could obtain 35 features from 58 features presented in [10]. Among them, we collected the number of words from the title, and from the description, we collected the number of words, the rate of unique words, the rate of words that are not stopwords, and the number of named entities. In addition to these, we collected the five most relevant topics collected from the descriptions, using the LDA [31] algorithm. The features related to the topics are the proximity of them to each video description. All of these attributes are extracted with Scikit-learn [90], Spacy [91], and NLTK [92] libraries.

Part of the features is related to subjectivity and sentiment polarity. Fernandes et al. [10] use the Pattern software to collect them. As this software does not support the Portuguese language, we use the Microsoft Azure cognitive services API [93] to fetch the Sentiment-based features. The polarity associated with a text sample can be {‘positive’, ‘neutral’, ‘negative’}; for the use of ML algorithms, we made the following conversion 1 for the positive polarity, −1 for negative polarity, and 0 for neutral. Likewise, the value of negative subjectivity is a real number that we multiplied by −1 before using the classifiers.

Using the publication date, it was also possible to obtain the day of the week when the video was published. We include two Boolean features to inform if the day is a Saturday or a Sunday. Table 5 exhibits the set with the 35 textual features.

### 6.4. Word Embeddings

Word embeddings are dense low-dimension real-valued vector representations for words that are learned from data. Their goal is to capture the semantics of words so that similar words have a similar representation in a vector space. Using word embeddings, one can expect not to rely on the attribute engineering stage, which often requires study and prior knowledge of the content to be predicted. In addition, if there is no knowledge about the texts to be analyzed, it is possible to obtain critical predictive features. As a counterpoint, we have the disadvantage of losing the interpretability of the features.

To collect the word embeddings from the title and descriptions, we use Facebook’s fastText [94] library for Python, which already comes with a pre-trained model for the Portuguese language. Their algorithm is based on the work of Piotr et al. [20] and Joulin et al. [95]. For each title and description, we first remove the stop words. Then, we run the fastText library and obtain a vector of 300 dimensions to the texts.

### 6.5. Classification

The popularity of content is the relationship between an individual item and the users who consume it. Popularity is represented by a metric that defines the number of users attracted by the content, reflecting the online community’s interest in this item [8]. Looking at the “most popular” videos or texts on the Web, the concept of popularity is intuitively understood. However, it is necessary to define objective metrics to compare two items and define which one is the most popular. Several measures point out which content attracts the most attention on the Web: the number of users willing to consume the item searched. In this work, we will use the number of views as a popularity metric.

The choice of machine learning models to conduct the classification task took into account the work done by Fernandes et al. [10] that selected the most used models in the researched literature. Furthermore, we group ML models into distance-based models (KNN), probabilistic models (Naive Bayes), ensemble models (Random Forest, AdaBoost), and function-based models (SVM and MLP). In this way, our choice tried to cover all these categories for comparison.

We use six classifiers to determine whether a video will become popular or not before its publication: KNN, Naive Bayes, SVM with a RBF, Random Forest, AdaBoost, and MLP. We performed five experiments to evaluate the effectiveness of these models. In the first experiment, we used only the 35 attributes obtained from Attribute Engineering as presented in Section 6.3. In the second, we used the vectors obtained with the fastText of the video descriptions, and, in the third, the predictive attributes were the word embeddings of the titles. In the last two experiments, we concatenate all the features. When combining the features engineered from the texts with the word embeddings, we reduce the dimension of the 300-dimension vector to 35 using the PCA and normalize all of them together. In this way, the 35 textual features do not lose representativeness given many features of the embeddings vectors.

As presented in most literature, we use a binary classification task in which a video is either popular or unpopular. We used the third quartile to define popularity in such a way that the goal was to find 25% of the most popular videos in our set. Our dataset has popular videos with 25% of the total and unpopular videos with 75%.

## 7. Results

In our experimental evaluation, we used six classifiers to analyze the results. The entire implementation was carried out in Python using the Scikit-learn library. After extracting the features, we have three datasets. The first has the 35 predictive textual attributes that we will call d_NLP. The dataset called d_Descriptions has the word embeddings of the video descriptions with 300 dimensions. Our third dataset brings the embeddings collected from the title vectors called d_Titles, also with 300 dimensions. The objectives of our experiments are to answer the following questions:
Does the video’s description contain information that a machine learning classifier can use to predict the popularity?How do the word embeddings features compare to attribute engineering in terms of the performance of the popularity forecast?

In all experiments, we follow a 10-fold cross-validation procedure to collect the predictive results. We balance the training set at each round of the cross-validation procedure with the Synthetic Minority Oversampling Technique (SMOTE) [96] algorithm implemented at imbalanced-learn [97] library. Similarly to Fernandes et al. [10], we performed GridSearch to find the best value for some hyperparameters for each ML classifier, namely, the number of trees for Random Forest and AdaBoost, the C trade-off parameter for SVM, the number of neighbors for KNN, and the number of hidden layers and their neurons.

To evaluate the predictive power of classifiers, we compute **accuracy**, **precision**, **recall**, and **F**-**measure**. Accuracy is defined in Equation (Equation 2). This metric is the complement of **Error Rate**, or incorrect classifications, presented in Equation (Equation 3). $\hat{f}$ is the classifier, $y_{i}$ the known class of $x_{i}$ and $\hat{f}\left(x_{i}\right)$ the predicted class, $\delta (y_{i},\hat{f}\left(x_{i}\right)=1$ if $y_{i}\neq \hat{f}\left(x_{i}\right)$ is true and 0, otherwise. With a problem of two classes, where one is popular content and the other unpopular, it is possible to present the Error Rate in a more understandable way as in Equation (Equation 4). FP are false positives, examples belonging to the unpopular class classified as popular and FN are false negatives, examples belonging to the popular class that are classified as unpopular. As in the case of popularity prediction, popular content is in the minority. The algorithms that classify the content as unpopular tend to have better accuracy. In this context, it is worse to have many false negatives.

Precision is defined in Equation (Equation 5), which presents the proportion of positive examples correctly classified among all those predicted as positive. Recall is defined in Equation (Equation 6), which corresponds to the hit rate in the positive class. In the Equations (Equation 5) and Equation 6, TP is the number of true positives, FP are the false positives, and FN is the number of false negatives.

The precision indicates the accuracy of the model, while the recall indicates completeness. Analyzing only the precision, it is not possible to know how many examples were not classified correctly. With the recall, it is not possible to find out how many examples were classified incorrectly. Thus, we usually compute the F-measure, which is the weighted harmonic mean of precision and recall. In Equation (Equation 7), *w* is the weight that weighs the importance of precision and recall. With weight 1, the degree of importance is the same for both metrics. The measure $F_{1}$ is presented in Equation (Equation 8).

### 7.1. Experimenting with Feature Engineered Textual Attributes

To answer the first question, we used d_NLP with the six classifiers to check the textual data popularity prediction performance. This experiment is the baseline of the analysis. The results are summarized in Table 6. The Random Forest (RF) classifier achieved the highest accuracy and F1-Score. In contrast, SVM showed high accuracy, but analyzing the accuracy, we found that the hit rate was satisfactory among those that the model claimed to be popular. When we looked at the very low recall, we noticed that several instances were FN cases. We calculated the importance of the features for the Random Forest model and listed the top-five in order of importance in Table 7. We found that the sentiment analysis features directly impact the popularity prediction. We still see the closeness to topic 2 of the LDA among the essential features. Below, we see the top ten words of the topic:
Top Words: [‘conká’, ‘arthur’, ‘gilberto’, ‘líder’, ‘karol’, ‘sarah’, ‘brothers’, ‘tieta’, ‘casa’, ‘bbb21’]

We found that these words refer to the reality show Big Brother Brasil 21, which started showing on 25 January 2021, and is very popular in Brazil. When checking the 20 most viewed videos in our dataset, only one (the 20th) does not refer to this program. It makes sense that this topic is one of the most relevant to popularity prediction with so many popular videos.

### 7.2. Experimenting with the Word Embeddings of the Descriptions

Using the dataset d_Descriptions, we observed that the MLP is the best model, but the accuracy decreased, and the result of the F1-Score decreased by approximately 10%. We also note that other models have suffered performance reductions. We found that attribute engineering better builds good predictive models when looking at the descriptions. The word embeddings probably capture much information contained within the description that is not related to the video popularity. Table 8 shows the results of the second experiment.

### 7.3. Experimenting with the Word Embeddings of the Title

In this experiment, we use the word embeddings obtained from the titles of the videos. The best model continues to be the RF, with an increase of 2.5% accuracy and 4.05% over the first experiment. This result contrasts with using the embeddings of the descriptions. In addition, we noticed that RF, SVM, KNN, and MLP increased performance. In the case of SVM, we had a significant improvement in all metrics. The good performance of MLP can be credited to the numerical nature of the embeddings that may provide features better suited to these models. Despite this, the AdaBoost and Naive Bayes models showed deterioration in the performance. As the 300 features represent the same title, they are not independent of each other, significantly affecting the Naive Bayes algorithm. However, for the AdaBoost algorithm, we expected an improvement close to that observed in the RF. The results can be seen in Table 9.

### 7.4. Joining Textual Engineered Attributes and Word Embeddings from the Titles

In this experiment, we concatenate the d_NLP and d_Titles datasets. Before the concatenation, we used the PCA to reduce the dimensions of d_Titles from 300 to 35, so the concatenated dataset was left with 70 predictive features. Random Forest remains the best model reaching an accuracy of 87%. Compared to the baseline, the SVM, RF, AdaBoost, and MLP models also improve. The SVM model achieved a precision of 88%, but the Recall was reduced, which means an increase in FN. The combination of traditional features with word embeddings provided a performance increase of 10% compared to baseline. Combining an effective engineering analysis of attributes with the representation offered by word embeddings provides us with an effective model for predicting popularity. The results are in Table 10.

### 7.5. Joining Textual Engineered Attributes and Word Embeddings

In this last experiment, we concatenated all three datasets. The dimensions of d_Titles and d_Descriptions have been reduced to 35 each, so the concatenated dataset has 105 predictive features. Although there was an increase in the performance compared to the baseline, the results found with this experiment were similar to the fourth experiment. One of the main conclusions is that the word embeddings of the description holds more generic information that do not help the classifiers. On the other hand, as seen before, the representation obtained by fastText from the video titles substantially improves the performance of the models. The results are in Table 11.

## 8. Discussion

This manuscript presented a review of the state-of-the-art and a real-world application to predict the popularity of content on the web using AI. Since it is not a trivial task, several strategies and models have been developed to determine which content attracts users’ attention on the Internet. Among them, selecting predictive attributes plays a central role in the performance of the models. We present a brief description of the theoretical foundation necessary to understand the theories, algorithms, methods, and results. We also defined a taxonomy for the classification of methods based on the tasks performed and according to attributes’ choice. The use of NLP to extract features, in general, provided the best results [10,14,15,16]. In addition to the textual information, the models also leverage metadata provided by the website that publishes the content. With the advance of DNN, it has become straightforward to extract attributes directly from the visual information of the images and videos.

The use of the popularity prediction for content optimization is still largely untapped and has enormous potential. The systems could suggest changes to the content using the predictors to see an upward trend in popularity. The primary beneficiaries of such an approach would be the creators of content that would increase the chances of attracting attention in the immensity of information that is the Internet [10]. We see that the classification algorithms worked better using textual attributes [13,16]. At the same time, the regressors obtained good results with metadata as attributes [22,23]. It is crucial to take this trend into account when developing new predictive models. Another venue that deserves further investigation is the use of different attributes and the extraction of features from multiple sources.

The selection of predictive attributes uses NLP methods extensively. We can mention the sentiment analysis task, NER, topic modeling with classic LDA, and the removal of *stopwords*. Among the ML algorithms, the *ensemble* methods proved to be more appropriate to the context of the popularity prediction. The ensemble methods successfully used by researchers were Random Forest, Bagging, and Gradient Boosting. In addition to these, traditional methods such as Naive Bayes, SVM, and KNN are often used as baselines. SVM still works as a basis for several methods that group the items according to the similarity of the evolution of popularity as in Trzcinski and Rokita [9].

After reviewing several previous works about the task of popularity prediction over web content, we can point out the importance of carefully choosing the attributes. The selection of attributes directly influences the performance of the predictive models, as we can see in Table 1 and Table 2. Still, defining attributes remains manual and with a closed goal of proving the hypotheses listed by the researchers. As a consequence, an exciting venue for further investigation is the automatic generation and selection of features with deep representation learning methods.

Predicting the popularity of web content has practical applications, for example, maximizing the return on marketing investment [8], proactively allocating network resources, fine-tuning them to future demands [9] and selecting the best content for a target audience [10,11]. Despite the development of research in this area and the sophisticated models presented, there are still several fields to be explored, such as content optimization, exploitation of data from social networks, and adaptation of real-world information to ML models.

## 9. Conclusions

As a case study, we analyzed six ML models to predict the popularity of videos from the most prominent streaming provider in Latin America. The dataset analyzed was obtained from Globoplay’s HTTP request logs. It has 9989 videos from the period from 25 January 2021 to 1 March 2021. The features used were primarily textual obtained from the titles and descriptions of the videos. We analyzed two different approaches to getting features: the first was to perform attribute engineering and obtain predictive attributes using NLP techniques; the second was representation learning, bringing features automatically through word embeddings. Finally, in the last experiments, we mixed the two approaches to see if there is an improvement when joining those features. The Random Forest model achieved the best results among those analyzed, obtaining an accuracy of 87% and F1-Score of 82% when used to combine the word embeddings of the titles and the attributes obtained through NLP. Among the features analyzed, we found that those associated with the polarity of words are the ones that most contribute to the popularity prediction. The description of the videos contributes significantly to popularity prediction when we use the feature engineering approach. Still, when we use fastText to get the word embeddings, we have had a substantial worsening in the performance of the classifiers.

As future work, we suggest associating textual features with visual features, for example, obtained through the thumbnails provided in the videos. In addition, it is possible to look for new ways to calculate the word embeddings of the descriptions trying to improve performance. Another exciting direction would be to understand the reason for reducing AdaBoost performance when using word embeddings from the video descriptions. Another exciting contribution would be to build a model that incorporates information from the offline world to predict the popularity of online content. There is clear evidence of the significant influence of real-world events on content popularity. For example, in 2020, 300 million people accessed the Internet for the first time due to the impacts caused by COVID-19 [3]. However, it is not easy to incorporate this information into predictive models. One suggestion would be to monitor social networks and news sites, incorporating specific attributes at the prediction time.

Finally, we emphasize that we made all predictions with features obtained before the videos are published. In this way, the built predictive model can be used for the correct dimensioning of the network infrastructures and assist in directing marketing costs.

## Figures and Tables

**Figure 1 sensors-21-07328-f001:**
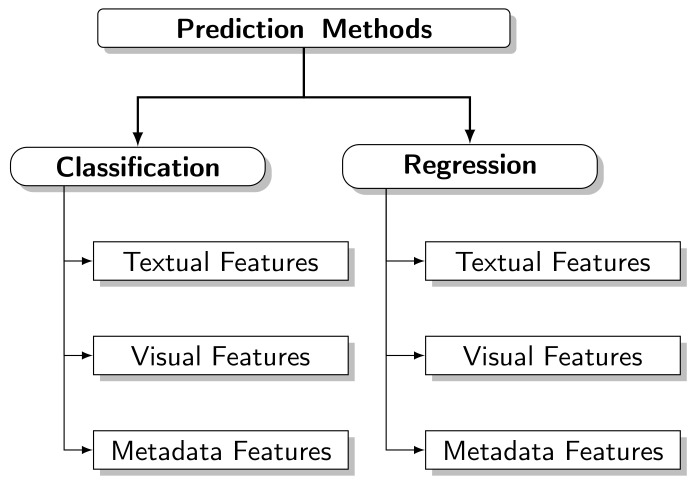
Taxonomy according to the prediction methods and attributes used.

**Figure 2 sensors-21-07328-f002:**
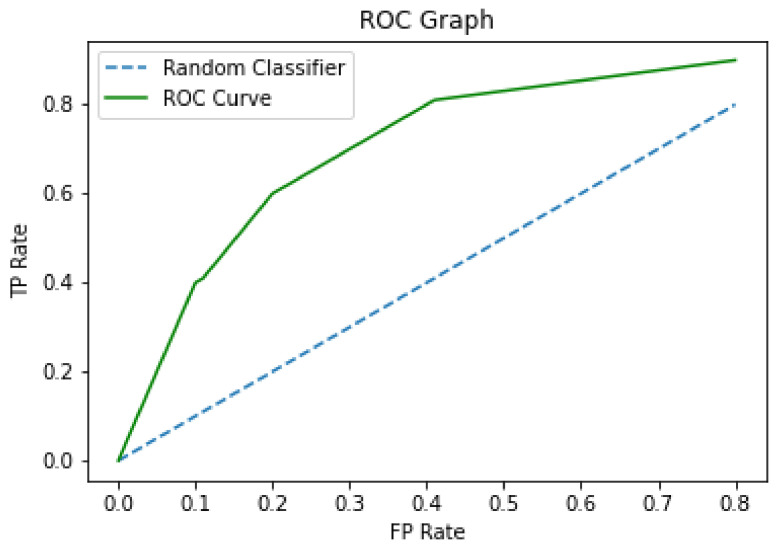
Example of the ROC curve.

**Figure 3 sensors-21-07328-f003:**
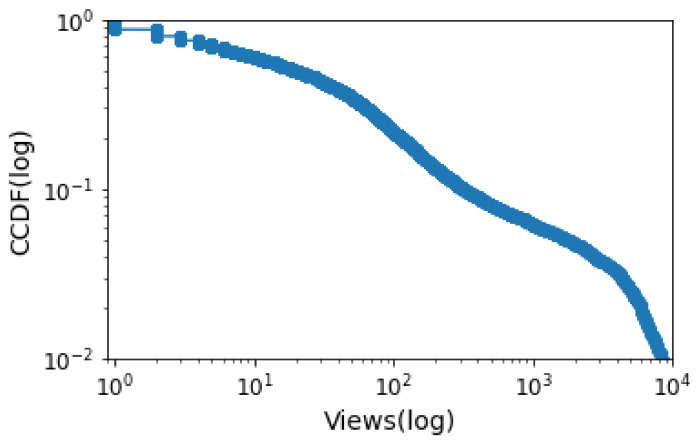
Complementary cumulative distribution function of number of views in log scale.

**Figure 4 sensors-21-07328-f004:**
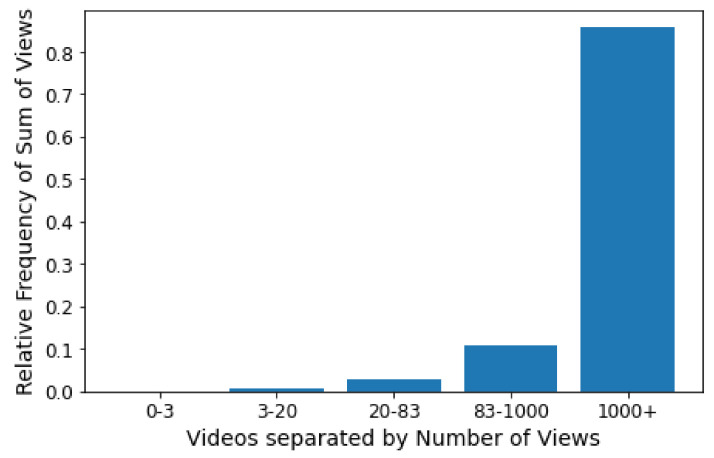
Percentage of total views separated by five classes of number of views.

**Figure 5 sensors-21-07328-f005:**
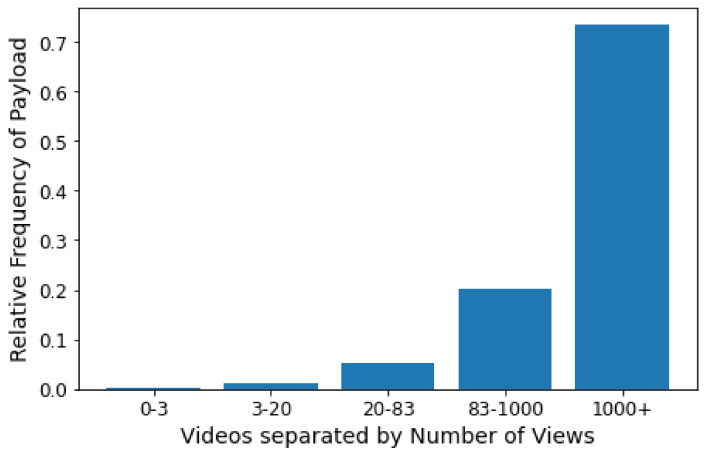
Percentage of total payload separated by five classes of number of views.

**Table 1 sensors-21-07328-t001:** Studies classified according to Taxonomy.

Prediction Task		Features				
C	R	Textual	Visual	Meta-Data	Content Type	References
	✓			✓	Videos and News	[22]
	✓	✓			Videos	[14]
	✓			✓	Videos	[23]
	✓		✓		Images	[21]
	✓		✓		Videos	[9]
	✓			✓	Videos	[39]
✓	✓	✓			News	[13]
✓		✓			News	[10]
✓	✓	✓			News	[15]
✓			✓		Videos	[11]
✓		✓			News	[16]
✓		✓			Videos	[40]

C = Classification R = Regression.

**Table 2 sensors-21-07328-t002:** Performance of models.

Best Performance				
Task	Model	Metric	Performance	References
R	LN model	RSE	graphic	[22]
R	Linear Regression	Spearman	0.8539	[14]
R	MRBF model	RSE	0.1723	[23]
R	SVR	Spearman	0.81	[21]
R	Popularity-SVR	Spearman	0.9413	[9]
R	CI Random Forest	$R^{2}$	0.8	[39]
C	Bagging	Accuracy	83.96%	[13]
C	Random Forest	AUC	0.73	[10]
C	AD Tree	AUC	0.837	[15]
C	Popularity-LRCN	Accuracy	0.7	[11]
C	Gradient Boosting	Accuracy	79%	[16]

C = Classification R = Regression.

**Table 3 sensors-21-07328-t003:** Features observed in literature.

Feature	References
Category	[10,13,15,16]
Author or Source	[13,15]
Title subjectivity	[10,16]
Content subjectivity score	[10,16]
Number of friends/followers of Author	[21]
Number of Named Entities	[13]
Number of keywords	[10,16,39]
Frequency of positive words	[14]
Frequency of negative words	[14]
Number of words in title	[10,15,16]
Number of words in content	[10,16]
HOG	[11,21]
GIST	[11,21]
Output of CaffeNet	[11,21]
Output of ResNet	[11,21]
Video’s length	[9]
Video’s resolution	[9]
HUE	[9]
Thumbnail contrast	[39]
Number of tweets/retweets	[13]
Number of Shares	[9,10,16]
Number of Views in the first day	[22,23,39]
Number of Views	[9,11,15,22]

Mainly features observed in literature about popularity prediction.

**Table 4 sensors-21-07328-t004:** Number of videos with corresponding percentage of total views and total payload.

Number of Views	Number of Videos	% Views	% Payload
0–3	2500	0.10	0.10
3–20	2564	0.60	1.10
20–83	2434	2.70	5.30
83–1000	1875	10.90	20.20
1000+	616	85.70	73.30

**Table 5 sensors-21-07328-t005:** Textual features collected from the title and the description of Globoplay.

Number	Feature	Number	Feature
1	Number of words of the title	19	Weekday is Saturday?
2	Number of words of the description	20	Weekday is Sunday?
3	Rate of unique words of the Description	21	Is Weekend?
4	Rate of non-stop words in the Description	22	Title Polarity
5	Rate of unique non stop words in the Description	23	Title Subjectivity
6	Average of word length in the Description	24	Description Polarity
7	Number of NER in the Description	25	Description Subjectivity
8	Topic LDA	26	Rate of Negative Words in Description
9	Closeness to LDA Topic 0	27	Rate of Positive words in the Description
10	Closeness to LDA Topic 1	28	Rate of Positive Words among non-neutral in the Description
11	Closeness to LDA Topic 2	29	Rate of Negative Words among non-neutral in the Description
12	Closeness to LDA Topic 3	30	Average of Negative Polarity among words in the Description
13	Closeness to LDA Topic 4	31	Maximum of Negative Polarity among words in the Description
14	Weekday is Monday?	32	Minimum Negative Polarity among words in the Description
15	Weekday is Tuesday?	33	Average of Positive Polarity among words in the Description
16	Weekday is Wednesday?	34	Maximum of Positive Polarity among words in the Description
17	Weekday is Thursday?	35	Minimum Positive Polarity among words in the Description
18	Weekday is Friday?	-	-

**Table 6 sensors-21-07328-t006:** Classification Results Features NLP.

Model	Precision	Recall	F1-Score	Accuracy
KNN	0.65	0.67	0.66	0.72
Naive Bayes	0.57	0.59	0.53	0.55
SVM	0.78	0.57	0.57	0.78
Random Forest	0.73	0.76	0.74	0.80
AdaBoost	0.68	0.68	0.68	0.76
MLP	0.71	0.73	0.72	0.78

**Table 7 sensors-21-07328-t007:** The five most important features in RF Model.

Feature	Importance
Avg polarity of Negative words	(1) 0.11636
Closeness to top 2 LDA topic	(2) 0.09072
Rate of Negative words	(3) 0.07067
Rate of Positive words	(4) 0.06947
Avg polarity of Positive words	(5) 0.05893

**Table 8 sensors-21-07328-t008:** Classification Results Embeddings Descriptions.

Model	Precision	Recall	F1-Score	Accuracy
KNN	0.59	0.61	0.61	0.52
Naive Bayes	0.56	0.56	0.42	0.43
SVM	0.64	0.68	0.65	0.71
Random Forest	0.63	0.65	0.64	0.72
AdaBoost	0.49	0.49	0.49	0.63
MLP	0.68	0.67	0.67	0.76

**Table 9 sensors-21-07328-t009:** Classification Results Embeddings Titles.

Model	Precision	Recall	F1-Score	Accuracy
KNN	0.70	0.75	0.70	0.74
Naive Bayes	0.59	0.59	0.45	0.45
SVM	0.74	0.77	0.75	0.80
Random Forest	0.77	0.77	0.77	0.82
AdaBoost	0.51	0.51	0.50	0.60
MLP	0.76	0.75	0.76	0.82

**Table 10 sensors-21-07328-t010:** Classification Results NLP + Titles.

Model	Precision	Recall	F1-Score	Accuracy
KNN	0.71	0.74	0.72	0.78
Naive Bayes	0.60	0.59	0.43	0.43
SVM	0.88	0.51	0.45	0.76
Random Forest	0.81	0.83	0.82	0.87
AdaBoost	0.77	0.80	0.78	0.83
MLP	0.75	0.75	0.75	0.81

**Table 11 sensors-21-07328-t011:** Classification Results Total Features.

Model	Precision	Recall	F1-Score	Accuracy
KNN	0.72	0.75	0.73	0.78
Naive Bayes	0.58	0.57	0.41	0.41
SVM	0.88	0.51	0.44	0.76
Random Forest	0.80	0.83	0.81	0.86
AdaBoost	0.77	0.80	0.78	0.83
MLP	0.74	0.74	0.74	0.80

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
