# Peer review of "Predicting Popularity of Video Streaming Services with Representation Learning: A Survey and a Real-World Case Study"

_sensors, 2021, doi:10.3390/s21217328_

Round 1

Reviewer 1 Report

In this paper, the authors conduct an extensive investigation of studies predicting the future popularity of video contents using various machine learning techniques. In addition, evaluation results are presented for effective models for accurate prediction.

  • As a survey paper, existing researches are well organized and summarized.
  • learning techniques is too long. Currently, general descriptions of these machine learning techniques are well documented in many literatures. You can reduce the amount of text written by adding a few references.
  • The use of capitalization in the axis titles of graphs is inconsistent in the writing format. Also, the font size is excessively large.
  • In judging the future popularity of a video by reflecting various features, it is necessary to provide more detailed data whether there is any genre bias in the video. In particular, when using machine learning by features for textual data and video images, it would be desirable if it was suggested in advance which features act as a major factor in determining popularity.

Author Response

Predicting Popularity of Video Streaming Services with Representation Learning: a survey and a real-world case study

Sidney Sá,  Antonio A. de A. Rocha, Aline Paes

Manuscript ID sensors-1395503, submitted to SENSORS (MDPI)

First of all, we thank you all for all the fruitful comments that certainly helped us to improve our paper. We have carefully considered all the remarks and suggestions made by the reviewers. We report below our answers regarding each remark raised by the reviewers and how we have addressed each remark in the revised version of the paper. It is worth mentioning that all the changes considered in the manuscript are marked in yellow, as suggested in the response letter, "such that changes can be easily viewed by the editors and reviewers". 

Responses to reviewers

=================================================================

Reviewer #1: In this paper, the authors conduct an extensive investigation of studies predicting the future popularity of video contents using various machine learning techniques. In addition, evaluation results are presented for effective models for accurate prediction.

As a survey paper, existing researches are well organized and summarized.

[Point 1] Learning techniques is too long. Currently, general descriptions of these machine learning techniques are well documented in many literatures. You can reduce the amount of text written by adding a few references.

[Response 1]

We appreciate the reviewer’s suggestion.  We reduced the details of the Machine Learning models in order to make it more direct and to facilitate the reading flow. We basically removed text from Section 2.1. 

[Point 2] The use of capitalization in the axis titles of graphs is inconsistent in the writing format. Also, the font size is excessively large.

[Response 2]

Thanks for noting. Figures 3, 4 and 5 have been improved. Both the capitalization and the font size adjusted properly.  

[Point 3] In judging the future popularity of a video by reflecting various features, it is necessary to provide more detailed data whether there is any genre bias in the video. In particular, when using machine learning by features for textual data and video images, it would be desirable if it was suggested in advance which features act as a major factor in determining popularity. 

[Response 3]

We appreciate the observations. We introduced a text part in Section 6.2 (Data Collection) where we report the types of videos in the dataset. We also present the features that most influence popularity in table 7 and are discussed throughout the text.

Reviewer 2 Report

This exciting review works on methods to predict the degree of satisfaction and acceptance (popularity) of streaming platforms using Machine Learning. However, some questions must be answered and need attention, namely:

1. The article is too long, even though it is a review article, causing the reader not to read the article continuously and fluently. The possibility of the article being reduced to the essentials should be considered; as it is, it is very hard to read and understand the advantages and disadvantages of each algorithm.
2. A table with acronyms must be included, given the number of abbreviations in the text.
3. Acronyms must be defined on the first occurrence and only once, as some appear defined more than once (see (ML), page 83 and page 134, (NLP) page 80 and page 265, etc.)

4. Flaws in the text, for example does not open "[" -> Top Words: 'conká', 'arthur', 'gilberto', 'leader', 'karol', 'sarah', 'brothers', 'tieta', 'home', 'bbb21'];
5. The keywords contain text from the model: "word embeddings (List three to ten relevant keywords specific to the article; yet reasonably common within the subject discipline.)".
6. Conclusions chapter "8. Conclusions" is essentially the discussion and should be called that way. The conclusion chapter should be summarized and have only the conclusions, not the discussion with references.
7. footnotes must be removed and placed in the references.

Author Response

Predicting Popularity of Video Streaming Services with Representation Learning: a survey and a real-world case study

Sidney Sá,  Antonio A. de A. Rocha, Aline Paes

Manuscript ID sensors-1395503, submitted to SENSORS (MDPI)

First of all, we thank you all for all the fruitful comments that certainly helped us to improve our paper. We have carefully considered all the remarks and suggestions made by the reviewers. We report below our answers regarding each remark raised by the reviewers and how we have addressed each remark in the revised version of the paper. It is worth mentioning that all the changes considered in the manuscript are marked in yellow, as suggested in the response letter, "such that changes can be easily viewed by the editors and reviewers". 

Responses to reviewers

=================================================================

Reviewer #2: This exciting review works on methods to predict the degree of satisfaction and acceptance (popularity) of streaming platforms using Machine Learning. However, some questions must be answered and need attention, namely:

[Point 1] The article is too long, even though it is a review article, causing the reader not to read the article continuously and fluently. The possibility of the article being reduced to the essentials should be considered; as it is, it is very hard to read and understand the advantages and disadvantages of each algorithm.

[Response 1]

We appreciate the reviewer’s suggestion.  We reduced the details of the Machine Learning models in order to make it more direct and to facilitate the reading flow, especially at Section 2.1. 

[Point 2] A table with acronyms must be included, given the number of abbreviations in the text.

[Response 2]

Thanks for the suggestion. We include a table with abbreviations and acronyms at the appropriate section (at the end of the paper), according to the MDPI template. 

[Point 3] Acronyms must be defined on the first occurrence and only once, as some appear defined more than once (see (ML), page 83 and page 134, (NLP) page 80 and page 265, etc.).

[Response 3]

We appreciate and totally agree with this observation. Thus, we considered this throughout the text. We defined acronyms always and only at the first time they appeared.

[Point 4] Flaws in the text, for example does not open "[" -> Top Words: 'conká', 'arthur', 'gilberto', 'leader', 'karol', 'sarah', 'brothers', 'tieta', 'home', 'bbb21'].

[Response 4]

Thanks for pointing that out. Those and others that were noted after submission  have been corrected in this revised version.

[Point 5] The keywords contain text from the model: "word embeddings (List three to ten relevant keywords specific to the article; yet reasonably common within the subject discipline.)".

[Response 5]

Thanks for pointing that out too. The text that did not belong to the keywords was removed.

[Point 6] Conclusions chapter "8. Conclusions" is essentially the discussion and should be called that way. The conclusion chapter should be summarized and have only the conclusions, not the discussion with references.

[Response 6]

We appreciate and agree with this suggestion. Thus, we decided to create a new section 8 (Discussion) to place the general dissertation about this work and moved the conclusion to section 9 in a more concise and dedicated section.

[Point 7] Footnotes must be removed and placed in the references.

[Response 7]

We appreciate the observation.  "Footnote Links'' have been added to the references with all the necessary information for each of them.

Reviewer 3 Report

In this paper, the authors identify the methods commonly used for the popularity prediction of videos in streaming services. The authors comment on the main features of each method. Two main approaches are identified based on the textual content of the videos: a) Feature engineering and b) Representation learning.

The paper is well organized and easy to follow. They give a comprehensive review of the prediction models and the fundamentals of each method which simplifies the subsequent sections since many concepts have been clearly described. They then use the six machine learning schemes to predict the popularity of unpublished videos, which I find very interesting. Using the prediction models they achieve a higher than 80% prediction accuracy using the Random Forest technique.

I believe that this paper could be of great use in the web popularity files prediction in future works.

I have only one question: The authors select eight commonly used machine learning methods to predict web content. This leads me to believe that there are other methods that can be used, even if they are not conventionally used, Is this correct? If so, why choose these eight methods in particular and no others? Could you give a brief explanation regarding this issue?

Author Response

Predicting Popularity of Video Streaming Services with Representation Learning: a survey and a real-world case study

Sidney Sá,  Antonio A. de A. Rocha, Aline Paes

Manuscript ID sensors-1395503, submitted to SENSORS (MDPI)

First of all, we thank you all for all the fruitful comments that certainly helped us to improve our paper. We have carefully considered all the remarks and suggestions made by the reviewers. We report below our answers regarding each remark raised by the reviewers and how we have addressed each remark in the revised version of the paper. It is worth mentioning that all the changes considered in the manuscript are marked in yellow, as suggested in the response letter, "such that changes can be easily viewed by the editors and reviewers". 

Responses to reviewers

 =================================================================

Reviewer #3: In this paper, the authors identify the methods commonly used for the popularity prediction of videos in streaming services. The authors comment on the main features of each method. Two main approaches are identified based on the textual content of the videos: a) Feature engineering and b) Representation learning.

The paper is well organized and easy to follow. They give a comprehensive review of the prediction models and the fundamentals of each method which simplifies the subsequent sections since many concepts have been clearly described. They then use the six machine learning schemes to predict the popularity of unpublished videos, which I find very interesting. Using the prediction models they achieve a higher than 80% prediction accuracy using the Random Forest technique.

I believe that this paper could be of great use in the web popularity files prediction in future works.

[Point 1] I have only one question: The authors select eight commonly used machine learning methods to predict web content. This leads me to believe that there are other methods that can be used, even if they are not conventionally used, Is this correct? If so, why choose these eight methods in particular and no others? Could you give a brief explanation regarding this issue?

[Response 1]

We appreciate the revision and comments. Regarding the posed questions we may say that following, and made it clear at the text.  The choice of machine learning models to perform the classification took into account the work done by Fernandes and used the most used models in the researched literature. Furthermore, we group ML models into distance-based models (KNN), probabilistic models (NB), ensemble-based models (RF, AdaBoost), and function-based models (SVM and MLP). In this way, our choice tried to cover all these categories for comparison. 

Round 2

Reviewer 2 Report

Thank you for your work.

All questions as been answered.

Author Response

Dear reviewer, thanks for the all the comments and suggestions. They were very valuable to improve the quality of our work.